# Robust total X-ray scattering workflow to study correlated motion of proteins in crystals

Steve P. Meisburger [1], David A. Case[2] & Nozomi Ando [1] ✉

The breathing motions of proteins are thought to play a critical role in function. However, current techniques to study key collective motions are limited to spectroscopy and computation. We present a high-resolution experimental approach based on the total scattering from protein crystals at room temperature (TS/RT-MX) that captures both structure and collective motions. To reveal the scattering signal from protein motions, we present a general workflow that enables robust subtraction of lattice disorder. The workflow introduces two methods: GOODVIBES, a detailed and refinable lattice disorder model based on the rigid-body vibrations of a crystalline elastic network; and DISCOBALL, an independent method of validation that estimates the displacement covariance between proteins in the lattice in real space. Here, we demonstrate the robustness of this workflow and further demonstrate how it can be interfaced with MD simulations towards obtaining high-resolution insight into functionally important protein motions.

Structural biology has seen remarkable advances in recent years with cryo-electron microscopy[1] and structure prediction[2,3]. However, protein crystallography remains the go-to method for obtaining structures at atomic resolution[4]. The technique is widely accessible and is still the dominant source of depositions in the Protein Data Bank. Crystallography also provides sub-angstrom coordinate precision and is therefore essential for benchmarking computational methods[5], such as simulations and structure prediction. Moreover, crystallography produces diffuse scattering[6], an untapped source of information on subtle protein motions that underlie processes such as allostery, catalysis, and signaling[7,8].

Crystal structures are often thought to represent static snapshots, but in fact, protein motions occur within the watery environment of crystals[9]. Diffuse scattering is a direct consequence of this motion, appearing as a structured, continuous signal in the background of diffraction images[10]. By studying the total scattering of a protein crystal (combining Bragg diffraction with diffuse scattering), crystallography has the potential to simultaneously provide a high-resolution average structure and information on correlated atomic displacements[5]. Until recently, diffuse scattering analysis was largely

considered intractable, but with advances in room-temperature data collection[11], the widespread availability of direct X-ray detectors[12], and new data processing software[13], it has now become feasible to routinely measure highly accurate diffuse scattering maps. However, although extensive efforts have been made in understanding protein diffuse scattering[7], a general workflow for utilizing this information has not yet been realized. In order to fulfill the promise of diffuse scattering in structural biology, it is essential to establish robust workflows for data processing, set standards for model-data agreement, and provide benchmark examples in the form of simulations and high-quality experimental data.

Our recent study of lysozyme in the triclinic (P1) space group showed that it is possible to account for the total scattering from a crystal entirely and self-consistently using physically-motivated atomistic models[13], and thus it can serve as a potential roadmap for establishing a standard workflow. A key advance from this study was the characterization of intense halo-like scattering around the Bragg peaks, which arise from correlated displacements of protein chains in different unit cells. Supercell simulations demonstrated that these correlations are long-ranged and consistent with phonon-like lattice

[1]Department of Chemistry and Chemical Biology, Cornell University, Ithaca, NY 14850, USA. [2]Department of Chemistry and Chemical Biology, Rutgers University, Piscataway, NJ 08854, USA. ✉e-mail: nozomi.ando@cornell.edu

vibrations, where proteins fluctuate about their average positions. To isolate internal motions of proteins from these external motions, it was first necessary to explain the halo features in the diffuse scattering map, which we achieved by fitting halos with a crystalline elastic network model treating the protein chain as a rigid body. Once the contribution of lattice disorder was accounted for, we were able to show that the remaining diffuse scattering signal and the B-factors from Bragg refinement were consistent with internal protein motions. In developing a workflow, the next step is to establish the accuracy and generality of lattice disorder models and develop tools for model validation. Ultimately, diffuse scattering analysis must provide new insight into biochemically relevant questions, and thus, it is also important that the workflow outputs data in a form that can be directly compared with atomistic modeling such as molecular dynamics (MD)[5].

Here, we introduce computational tools and demonstrate a robust workflow to isolate the internal motion signal from total scattering data (Fig. 1). To model lattice disorder, we present GOODVIBES, a general crystalline elastic network model and optimization routine. This method allows for multiple rigid bodies per unit cell, as found in high-symmetry space groups, and accounts for symmetry in its parameterization of the elastic network. To validate the lattice disorder model, we present an independent method called DISCOBALL, which estimates rigid-body displacement covariances for pairs of protein chains in the crystal by deconvolution of the 3D pair distribution function or 3D-ΔPDF (the Fourier transform of the diffuse scattering intensities). Using simulated data and experimental datasets from three lysozyme polymorphs, we show that GOODVIBES and DISCOBALL in combination can be used to accurately model lattice disorder scattering. Finally, we show that the signal from internal protein motion can be recovered and compared quantitatively with crystalline MD simulations. The demonstration of a general workflow for diffuse scattering analysis lays the groundwork towards obtaining atomistic insight into correlated protein motions from experimental data.

## Results

### A robust workflow to isolate internal motion signal from total scattering data

An overview of a general workflow for the analysis of total scattering from protein crystals is shown in Fig. 1. The first step begins with data collection and reduction, the procedures for which we previously

demonstrated[13]. Briefly, diffraction images are acquired from protein crystals at room temperature as well as from the background scattering using a room-temperature macromolecular X-ray crystallography (RT-MX) setup. Data reduction can then be performed with the *mdx* software library, which we introduced previously[13]. Measurement of the background scattering and a careful scaling procedure (such as that implemented in *mdx-lib*) allow for a high-quality reconstruction of the total scattering (TS) on an absolute scale (electron units), which includes Bragg peak intensities and a three-dimensional diffuse scattering map. The time-averaged electron density of the unit cell is then determined by conventional structure refinement of the Bragg data, along with the mean atomic coordinates and atomic displacement parameters (ADPs) or B-factors. The ADPs represent the motion of each atom, while the diffuse scattering map contains information on how these motions are correlated.

In our previous work[13], we showed that correlated motions arise from two sources in protein crystals: the motion of atoms within a protein (internal motion) and deviations from the ideal arrangement of proteins in the crystal (external motion or lattice disorder). Lattice disorder tends to be long-ranged, and therefore it produces characteristic halos around the Bragg peaks, while correlations from internal motion are mostly short-ranged and produce smoothly varying, cloudy patterns. Although the two types of signals have distinct appearances, they cannot be simply separated in reciprocal space because the halo features are remarkably broad and overlap significantly with the much weaker and more nuanced cloudy pattern. Therefore, we aimed to develop additional tools based on a physical model of lattice disorder to subtract its contribution from the diffuse scattering map and the ADPs.

First, it was necessary to derive a general physical model of lattice disorder that can be refined to fit the diffuse halos (Fig. 1, green box). Based on the success of parameterized crystalline elastic network models in the case of triclinic lysozyme[13], we chose to extend those techniques to arbitrary space groups with multiple rigid bodies per unit cell. We call the parameterization and refinement method GOODVIBES for General Optimization Of Diffuse halos from VIBrational Elastic network Simulations. The GOODVIBES model for lattice disorder represents proteins as rigid bodies arranged in a supercell with periodic boundary conditions (Fig. 2a). The size of the supercell is chosen to be large enough to account for the long-ranged correlations

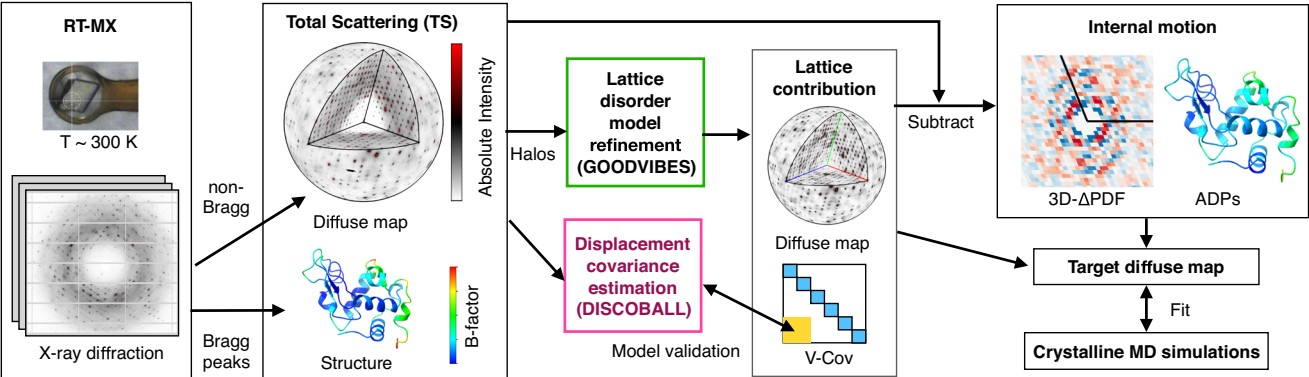

**Fig. 1 | Workflow to measure and interpret protein correlated motion using X-ray crystallography.** First, X-ray diffraction images are acquired from protein crystals at room temperature (RT-MX). The Bragg peaks and continuous scattering are processed separately to obtain the protein structure and a three-dimensional map of diffuse scattering on an absolute intensity scale (electron units). The structure includes mean atomic positions and atomic displacement parameters (ADPs or B-factors) that quantify motion, and the pattern of diffuse scattering depends on how motions are correlated. To separate the internal and external (rigid-body) protein motions, a physical model of lattice disorder is refined to the intense diffuse halo features (GOODVIBES), and the lattice contribution to the

diffuse map and variance-covariance matrix of rigid-body motion (V-Cov) are simulated. In parallel, a model-free analysis is performed to estimate displacement covariances (DISCOBALL) and validate the off-diagonal elements of the simulated lattice V-Cov (yellow shading). The lattice contribution to the diffuse map is subtracted and the residual diffuse scattering is sorted by inter-atomic vector using a Fourier transform (3D-ΔPDF). Similarly, the internal ADPs are found by subtracting the lattice contribution (diagonal blocks of V-Cov, blue shading). The internal motion signal can be interpreted by various models. To match crystal simulations, a target diffuse map can be created using GOODVIBES to add back external motions that are consistent with the specific supercell used by the simulations.

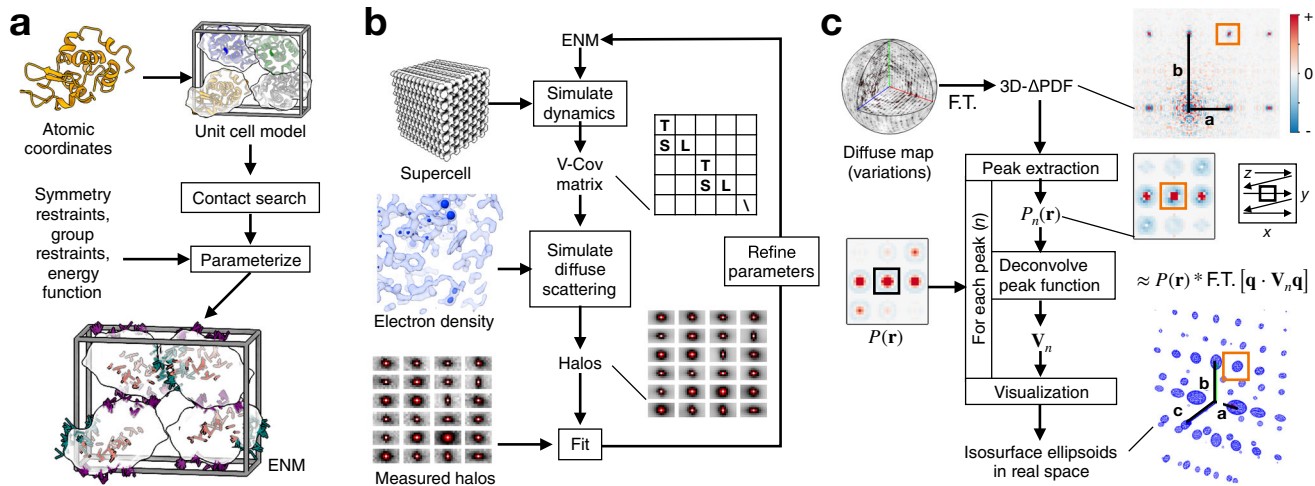

**Fig. 2 | Lattice disorder modeling with GOODVIBES and DISCOBALL.**
**a** Preparation of a rigid-body elastic network model (ENM) for GOODVIBES refinement illustrated using lysozyme in the P2₁2₁2₁ space group. Rigid bodies (white surfaces) are joined by springs with refineable energy functions. The parameterization step enforces space group symmetry and groups springs to avoid overfitting. In this example, springs are grouped according to unique protein-protein interface (sticks colored orange, teal, and purple). **b** GOODVIBES refinement of the ENM to fit diffuse halos. At each iteration, the variance-covariance matrix of rigid body motion (V-Cov) is computed from the ENM by simulating the thermally-excited vibrations of a large supercell with periodic boundary conditions. The halo scattering profiles are simulated from V-Cov and the electron density of

the asymmetric unit, and ENM parameters are refined to improve the fit.
**c**, DISCOBALL algorithm to estimate correlated rigid-body motion of proteins. Peaks are extracted from the 3D-ΔPDF, the Fourier transform (F.T.) of the diffuse map. Each peak, $P_n$, is assumed to be the convolution (*) of the Patterson origin peak, $P(r)$, and a function that depends on the average joint-ADP $V_n$ of proteins related by a lattice translation operator (integer multiples of lattice vectors). The joint-ADPs are estimated by deconvolution. Joint-ADPs are visualized as iso-probability ellipsoids in real space (blue mesh) to illustrate the anisotropy of each ADP and the overall decay of correlations with protein-protein distance in the crystal (vector from the origin).

implied by halo features in the experimental map. Rigid-body motion is enforced by adopting a generalized coordinate system for small displacements, which is a reasonable assumption for the subtle structural fluctuations we expect. The coordinate system, which is identical to that used in translation, libration, screw-axis (TLS) refinement, assigns each rigid body six degrees of freedom representing small translations and rotations. The close contacts between proteins in the lattice are modeled using a network of inter-molecular springs, whose functional form and strength can be tuned. The dynamics of the system are computed using Newtonian mechanics and equipartition of thermal energy (see Methods).

A key advantage of the GOODVIBES model is that both the displacement covariance matrix and the predicted diffuse scattering can be calculated analytically with minimal computational cost (see Methods for details). This allows for a closed refinement loop, where the parameters in the model are optimized to improve the fit between simulated and measured diffuse scattering (Fig. 2b). We demonstrated this approach previously with triclinic lysozyme[13]. Here, we extend the modeling to unit cells with more than one protein chain with a parameterization method that accounts for space group symmetry by forcing springs related to a symmetry operator to have the same parameters.

Second, we required an independent method to test whether the model for lattice disorder is sufficiently accurate (Fig. 1, magenta box). We developed a model-free approach, DISCOBALL, for DISplacement COVariance Between ALl Lattice neighbors. DISCOBALL was inspired by features observed in the 3D-ΔPDF of triclinic lysozyme[13]. While the short-range contributions of internal protein motions are focused near the origin of the 3D-ΔPDF, sharp peaks are present at the lattice nodes as a direct consequence of halo features centered at reciprocal lattice nodes in the diffuse map (the Fourier transform of a lattice is another lattice). These sharp 3D-ΔPDF peaks have a simple physical interpretation: they arise from correlated displacements of atoms separated by a lattice translation. The position of the peak corresponds to a vector between unit cells in the

crystal, and all atoms offset by that vector contribute to the peak if their motions are correlated.

The mathematical form of the 3D-ΔPDF peaks can then be derived by considering the diffuse scattering arising from correlated motions of rigid bodies that are separated by unit-cell vectors and then taking the Fourier transform (see Methods). For simplicity, we may assume that translational motions of unit cells are the dominant contribution as rotational motions are only likely to be correlated between nearest neighbors. This can then be generalized to the case where there are multiple rigid bodies in a unit cell that are related by space group symmetry (see Methods). The mathematical form of the 3D-ΔPDF peaks can be interpreted as a filtered version of the origin peak in the Patterson map (the autocorrelation of the electron density), where the filtering function depends on the displacement covariance, or joint-ADP, of pairs of proteins (see Methods). From a signal processing point of view, the 3D-ΔPDF peaks are exactly a convolution of the Patterson origin peak with the filtering function. Using the known Patterson function (e.g., from Fourier transform of the Bragg intensities), the joint-ADPs can be recovered by deconvolution (Fig. 2c).

To test the DISCOBALL method on a realistic dataset, we simulated the diffuse scattering from a GOODVIBES model for triclinic lysozyme, added Gaussian random noise comparable to the experimental map, and computed the 3D-ΔPDF. We found that DISCOBALL was able to recover both the magnitude and shape of the joint-ADPs (Supplementary Fig. 1). For a more quantitative test, we derived two statistical metrics for the similarity between joint-ADPs: the total covariance (trace of the joint-ADPs, Eq. (22) in Methods) and the anisotropic covariances (elements of the joint-ADP with isotropic part subtracted, Eq. (23) in Methods). By comparing these metrics with corresponding ground truth values, we find that DISCOBALL is also able to recover joint-ADP magnitudes and shapes with high fidelity (Pearson correlation of 0.999 and 0.988, for isotropic and anisotropic metrics respectively).

After lattice dynamics are adequately modeled, the final step in the workflow is to subtract its contribution and obtain only the signal

from internal protein motion. Previously, a major obstacle to this approach was the potential for errors in the refined structure model to manifest in the diffuse scattering simulation, which would then produce even more significant errors in the subtracted signal[13]. Here, we have addressed this concern by using experimental structure factor amplitudes in the GOODVIBES simulation. This approach succeeds because measurement precision is usually much better than model-data agreement in protein crystallography[14] (for examples studied here, $R_{pim}$ ~1% vs. $R_{work}$ ~10%). When investigating the accuracy of our Bragg data, we unexpectedly found that intensities were systematically suppressed in an intensity- and resolution-dependent manner. Interestingly, this behavior is reminiscent of dynamical scattering (extinction), and we corrected for it using a method developed for small-molecule crystallography (Supplementary Fig. 2). We also subtract the ADPs according to GOODVIBES for each of the atoms in the refined structure. These residual ADPs more accurately represent the internal motions of the protein.

Our workflow provides the opportunity to compare experimental data with atomistic simulations. For greatest accuracy, all-atom MD is needed to reproduce both the correlations internal to the protein as well as its surrounding solvent[7]. All-atom MD can also account for effects of crystal packing by using a supercell with periodic boundary conditions[15]. Previously, such simulations in the crystalline state have been found to reproduce diffuse scattering data with reasonable accuracy[16] including halos if increasingly large supercells are used[13]. Here, we asked whether small-scale simulations, of one or a few unit cells, might be compared with the subtracted maps in order to maximize the proportion of the scattering signal from internal motion. A potential issue with such comparisons is that small-scale simulations still include external protein motion to some extent. Since GOODVIBES also involves a periodic supercell, we reasoned it could estimate the amount of external motion that ought to occur in an MD simulation of arbitrary size. This allowed us to create a target diffuse map for direct comparison with any particular crystalline MD simulation.

The procedure for creating target diffuse maps is described in Methods. Briefly, we first fit a GOODVIBES model and subtract its scattering contribution from the experimental data, as described above. The subtracted map is on the same Miller index grid as the experimental data. To create the target map, it is first necessary to interpolate the subtracted map onto the Miller index grid of the MD simulation (e.g. for a single unit cell, points with integer Miller indices). The subtracted maps are noisy, both from photon counting statistics as well as errors in the simulated intensities, and they also contain gaps at Bragg peak locations. To account for these errors and gaps during interpolation, we take advantage of the expected smoothness of diffuse scattering from internal motions. This smoothness comes about because inter-atomic correlations are restricted to a maximum distance defined by the protein envelope, i.e. the diffuse scattering is a band-limited function. For a smoothing interpolant we chose an error-weighted Savitzky-Golay filter[17] with a kernel corresponding to the reciprocal dimensions of the protein. The effect is similar to a low-pass filter, but it also handles outliers, missing data, and error propagation (see Methods for details). Finally, to produce the target map, the diffuse scattering from external motions is simulated using GOODVIBES with the MD supercell and added to the subtracted, interpolated map. The target map can then be compared directly with the MD simulation on an absolute scale.

### Application of GOODVIBES and DISCOBALL to experimental datasets from lysozyme polymorphs

To test the workflow on experimental data, we applied it to TS/RT-MX datasets from three lysozyme polymorphs: triclinic lysozyme reported previously[13,18,19] and new datasets from orthorhombic ($P2_12_12_1$) and tetragonal ($P4_32_12$) crystals (Fig. 3a, top row). Because protein crystals are highly susceptible to radiation damage at room temperature, the

total dose was distributed by collecting narrow wedges from multiple locations on each crystal and from multiple crystals when necessary to obtain a complete dataset (Fig. 3a, middle row and Supplementary Table 1). The maximum tolerable dose of ~65 kGy[13] was set conservatively by monitoring the decay of the Wilson B-factor as well as evidence for disulfide bond reduction in the electron density maps (Supplementary Fig. 3). Crystals from each polymorph diffracted to atomic resolution allowing anisotropic ADPs to be refined for each atom, and stereochemically high-quality structures with low R-factors were obtained (Supplementary Table 2).

Next, the maps of diffuse scattering from orthorhombic and tetragonal lysozyme were reconstructed from the diffraction images (Fig. 3a, bottom row, and Supplementary Table 3). Low crystal mosaicity and fine phi slicing of 0.1 degree allowed for sampling on a fine grid, comparable to the triclinic map reported previously[13,19]. All three datasets contain strong halo features at the Bragg peak locations (Fig. 3b, bottom row). To model these halo features, a subset of 400 halos was selected for each dataset (blue rectangles in Fig. 3b), and GOODVIBES was used to parameterize and optimize the spring constants of a crystalline elastic network model (Fig. 3b, top row). After refinement, the diffuse map was simulated throughout reciprocal space. For each of the three polymorphs, GOODVIBES simulations show remarkable agreement with the observed diffuse patterns (Fig. 3b, middle row). The simulations reproduce the standard deviation of intensity in all resolution bins (Fig. 3c, upper axes), and the correlation coefficient in each resolution shell is close to the theoretical limit allowed by signal-to-noise of the data (CC vs. CC*, Fig. 3c, lower axes).

To quantify how well the GOODVIBES models explain lattice disorder in each lysozyme polymorph, we performed a complementary DISCOBALL analysis. The diffuse maps were pre-processed to remove the isotropic scattering and fill in missing voxels (such as at Bragg peak locations), and the 3D-ΔPDFs were computed using a Fourier transform as described in Methods. The 3D-ΔPDFs displayed sharp peaks at each lattice point that become weaker with distance from the origin peak (Supplementary Fig. 4). The joint-ADPs between neighboring chains were obtained by deconvolution of the Patterson origin peak as described in Methods. The number of unique joint-ADPs that can be determined by DISCOBALL analysis depends on the number of peaks in the ASU of the 3D-ΔPDF, which is a function of space group symmetry, unit cell size, and reciprocal space sampling (Supplementary Table 3). The GOODVIBES model, in contrast, predicts displacement covariances for each pair of proteins in the crystal (the number is comparable for the three polymorphs). Thus, for the triclinic case, DISCOBALL can cross-check all of the joint-ADPs predicted by GOODVIBES model, while for orthorhombic and tetragonal only a subset can be compared.

Joint-ADPs from DISCOBALL and GOODVIBES analyses are compared in Fig. 4a. Overall, the total displacement covariance shows a similar decay profile for all GOODVIBES models (Fig. 4a, green symbols). A similar overall behavior is expected since the models have the same essential physics (lattice vibrations). The total covariances are negative at large distances. This can be understood as a consequence of the finite supercell size. Since the supercell's center of mass is fixed in space, correlated motion at short distance must be canceled by anti-correlated motion at large distance. The covariances from DISCOBALL analysis show a similar overall decay (Fig. 4a, blue symbols), however the decay approaches closer to zero than the GOODVIBES predictions. In DISCOBALL analysis, the extra long distance information comes from the interpolation of the diffuse map at the Bragg peak locations. For a quantitative comparison, the different asymptotic behaviors at large distances were taken into account by fitting a straight line to the scatter plot of DISCOBALL vs. GOODVIBES total covariances (Fig. 4b). As expected, the line does not pass through the origin (it falls below $y = x$). The agreement between the two methods was quantified using the Pearson correlation coefficient ($r$) corresponding to the linear fit.

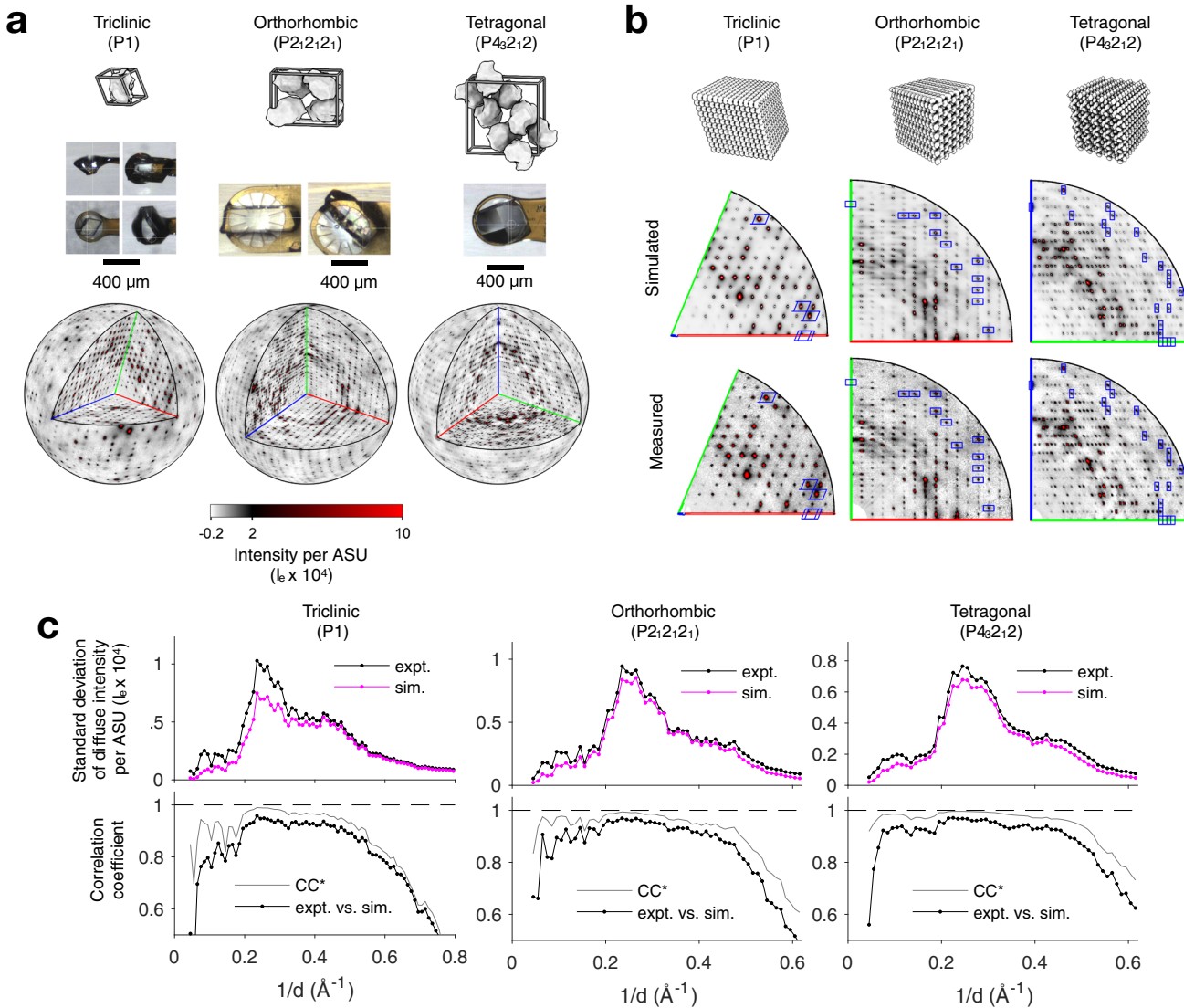

**Fig. 3 | Application of GOODVIBES to experimental datasets from lysozyme polymorphs. a** TS/RT-MX datasets from lysozyme crystallized in triclinic, orthorhombic, and tetragonal space groups. The unit cells contain one, four, and eight symmetry-related chains, respectively (top row). X-ray diffraction data were acquired at room temperature from multiple large crystals to achieve high signal-to-noise for diffuse mapping (photographs of mounted crystals, middle row). Data were processed to produce three-dimensional maps of diffuse scattering. In the bottom row, the variational component of intensity (total minus isotropic) in electron units ($I_e$) per asymmetric unit (ASU) is shown on a spherical surface at 2 Å resolution, with the positive octant removed to show three central sections. The axes [*h*,0,0], [0,*k*,0], and [0,0,*l*], are colored red, green, and blue, respectively. **b** GOODVIBES was used to fit a lattice disorder model to each diffuse scattering dataset using supercells shown in the top row. After refinement to a subset of intense halos, the diffuse scattering from lattice disorder was simulated throughout reciprocal space. For all three datasets, GOODVIBES reproduces essential features of the diffuse scattering, as seen for example in central sections (middle vs. bottom row). Blue boxes surround halos included in the fit. The black line is at 2 Å resolution, and axes are drawn as in (**a**). **c** For all three datasets, lattice disorder accounts for most of the standard deviation of the intensity in each resolution bin (top row of plots), as well as the precise pattern of intensities as judged by the Pearson correlation coefficient in each resolution shell (bottom row of plots). The correlation approaches the theoretical limit given signal-to-noise of the measurement[13,57] (CC*, black vs. gray lines in the bottom row of plots).

Based on this quantitative analysis, the GOODVIBES models do an excellent job describing the total covariance, with *r* values in excess of 0.995 (Fig. 4b).

We also tested the ability of GOODVIBES models to account for the anisotropic components of the joint-ADPs. In all three polymorphs, the GOODVIBES models and DISCOBALL analysis show a strong correlation (Fig. 4c). Interestingly, orthorhombic and tetragonal lysozyme have similar magnitudes of the total correlation (Fig. 4b, middle vs. bottom panels), but the spread of anisotropic components for orthorhombic is approximately double that for tetragonal (Fig. 4c, middle vs. bottom panels).

To better understand how crystal mechanics give rise to such strong anisotropy in the case of the orthorhombic crystal, we overlaid

a diagram of the crystal lattice with the joint-ADPs shown as isosurface ellipsoids (Fig. 4d). The strength and directionality of the mechanical couplings between proteins can be visualized by the size and shape of the ellipsoids (which represent the magnitude and anisotropy of their joint-ADPs). The couplings relative to the asymmetric unit (Fig. 4d, yellow protein) are strong in the ±**a**, ±**b**, and +**c** directions (Fig. 4d, double arrows). In comparison, couplings are weak in the −**c** direction across the continuous solvent channels running parallel to the **a** axis (Fig. 4d, purple regions). In addition, the joint-ADPs are elongated relative to the protein-protein vector in the ±**b** directions, where the sequence of direct interactions is unbroken by solvent channels. Thus, the anisotropy of joint-ADPs can be understood as a consequence of the topology of solvent channels and crystal contacts.

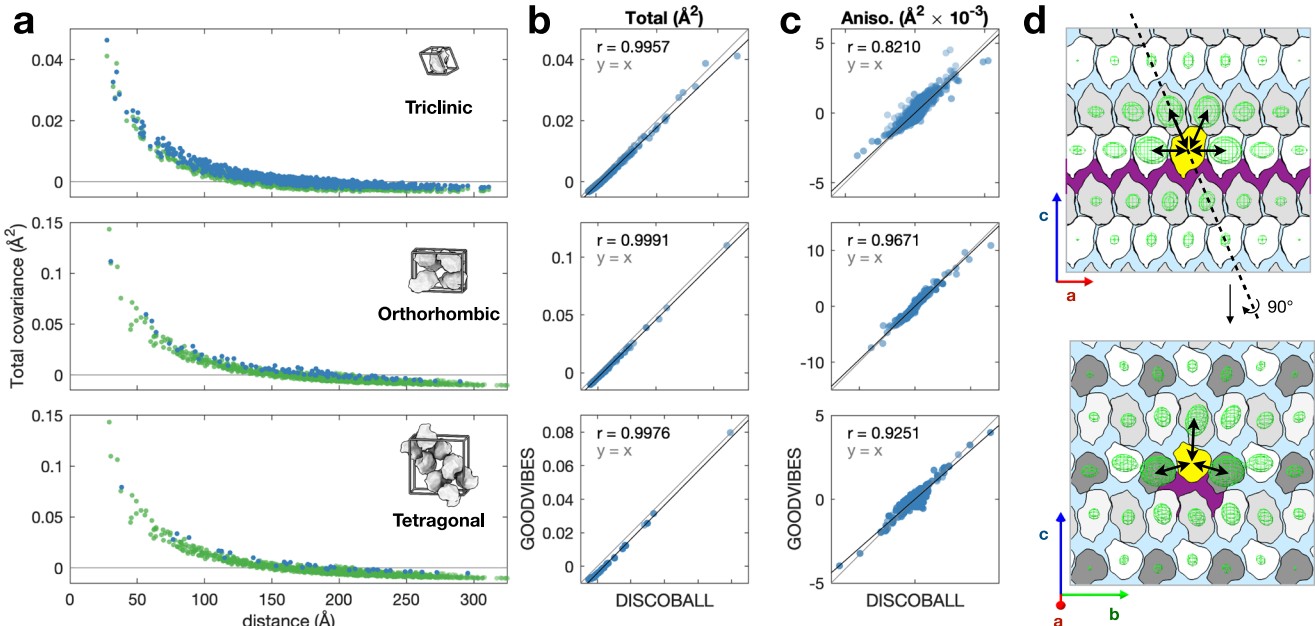

**Fig. 4 | DISCOBALL analysis and validation of lysozyme polymorph datasets and insight into crystal mechanics.** Displacement covariances matrices (joint-ADPs) were estimated for triclinic, orthorhombic, and tetragonal polymorphs (top to bottom in **a**–**c**) and split into total and anisotropic components for further analysis (defined in Methods). **a** The total covariance from DISCOBALL (blue points) and GOODVIBES (green points) decays with distance between protein molecules, as expected for an elastic crystal. **b** The GOODVIBES fit shows improved agreement to the total covariances for all polymorphs (Pearson correlations, *r*, inset). **c** Validation of GOODVIBES according to the anisotropic components. Anisotropy is especially significant in the orthorhombic crystal (middle panel). **d** Sections through the orthorhombic crystal are overlaid with isosurface representations of the joint-ADPs from the GOODVIBES model (green mesh) for each protein relative to the asymmetric unit (yellow shading). Correlated motion between nearest neighbors is strong (black arrows) except across the continuous solvent channel running parallel to the *a* axis (purple shading).

After validating the model of lattice disorder, the next step is to subtract the lattice disorder component and examine the remainder. First, we examined the lattice contributions to the total ADPs. Although GOODVIBES models are refined to diffuse scattering data, they directly predict TLS matrices (Supplementary Table 3), which in turn can be converted into anisotropic ADPs for each atom (see Methods). We compared these model ADPs directly with the individual anisotropic ADPs obtained from structure refinement. For visual clarity, we grouped backbone atoms by residue and averaged their equivalent isotropic B-factors (Eq. (6) in Methods). These per-residue B-factors are plotted for each lysozyme polymorph in Fig. 5a. A distinct pattern of variation is observed in each polymorph (Fig. 5a, black line and points). The median backbone B-factor for triclinic lysozyme is 8.24 Å$^2$, approximately half of the median backbone B-factors of orthorhombic and tetragonal (17.66 and 17.74 Å$^2$ respectively).

The lattice disorder ADPs obtained from GOODVIBES include both overall translational motion, which is the same for all atoms (Fig. 5a, light blue region), and rotational motion, which causes the ADP to vary depending on atomic position (Fig. 5a, dark blue region). The translational lattice motion varies between polymorphs, with equivalent B-factors of 3.06, 8.08, and 8.84 Å$^2$ for triclinic, orthorhombic, and tetragonal, respectively (Fig. 5a, light blue region). As noted above, the orthorhombic and tetragonal forms have median B-factors that are approximately twice that of triclinic. The same trend is observed in the lattice components, suggesting that lattice disorder is largely responsible for the B-factors of the most ordered atoms in a protein crystal.

The contribution of internal protein motion to diffuse scattering can be visualized readily in the 3D-ΔPDF, since the signal from short-ranged correlations is concentrated near the origin. We computed the 3D-ΔPDF for the experimental diffuse maps and GOODVIBES simulations (Fig. 5b). In all polymorphs, variations in the measured 3D-ΔPDFs agree with the GOODVIBES simulations for *d*>10 Å, but the variations are stronger in the region *d*<10 Å where internal motions are expected to contribute[13]. The residual 3D-ΔPDF (total minus simulation) decays rapidly with distance from the origin in a similar manner for each polymorph (Fig. 5c). These features are consistent with the hypothesis that the residual signal derives from internal protein motion.

## Application of GOODVIBES to match experimental data to MD simulations

To demonstrate the final step in our workflow, a single unit-cell MD simulation of tetragonal lysozyme was performed (Supplementary Fig. 5), and the diffuse scattering was calculated from the snapshots. Because the simulation had periodic boundary conditions, the diffuse scattering is defined only at the points in reciprocal space with integer Miller indices. For comparison, we re-processed the tetragonal dataset to produce a target map, which includes external motion of the proteins that are compatible with the MD supercell (Supplementary Fig. 6a) using the method described earlier. Overall, the MD simulation and the target maps have a remarkably similar appearance (Fig. 6a). They also show quantitative agreement (Fig. 6b). Both include a strong isotropic ring of scattering at ~3 Å resolution (Fig. 6b, top panel). The MD simulation also reproduces the magnitude of scattering variations in each resolution shell, including a distinctive shoulder at ~2 Å resolution (Fig. 6b, middle panel). A comparison of MD and the target map's internal and external components provides compelling evidence that external protein motions alone cannot explain diffuse scattering (Supplementary Fig. 6b) and furthermore, that the atomic fluctuations are captured well by MD.

Previous comparisons between MD and experiment have relied on correlation coefficients computed within resolution shells[13,16,20]. We applied a similar approach using the target diffuse map (Fig. 6b, bottom panel). MD agrees best at low resolution (CC ~0.8 at 10 Å), and the agreement is somewhat reduced at higher resolution (CC ~0.5 at 2 Å).

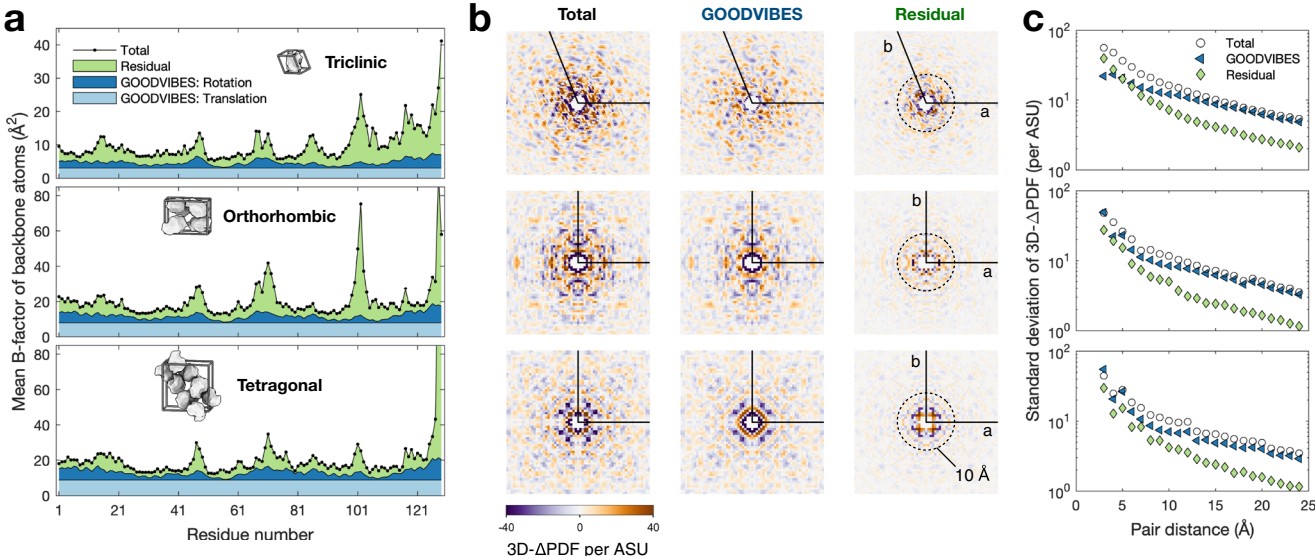

**Fig. 5 | Contribution of internal protein motion to ADPs and diffuse scattering.**
**a** ADPs of backbone atoms for each lysozyme polymorph (top to bottom: triclinic, orthorhombic, and tetragonal). For clear visualization, the full set of ADPs was reduced to a mean equivalent isotropic B-factor for each residue. Using GOOD-VIBES, the total B-factor (black line and symbols) is decomposed into rigid-body motion due to lattice disorder (light and dark blue shading) and the residual attributed to internal protein motion (green shading). As described in Methods, the B-factors from rigid-body motion were further decomposed into translations (light blue shading) and rotations (dark blue shading), which vary among residues depending on their distance from the rotation center. The contribution of lattice disorder to B-factors is significant in all three polymorphs, and the residual B-factors have a similar overall magnitude (note that the y-axis scale is different for

the triclinic polymorph). **b** 3D-ΔPDFs computed from the experimental and simulated diffuse maps for each polymorph (arranged top to bottom, as in (**a**)). Central sections are shown in the **a**–**b** plane for the experimental map (left panels), GOODVIBES simulation (middle panels), and the residual after subtraction (right panels), on the same intensity scale normalized per asymmetric unit (ASU). The region near the origin is shown (dashed circle has a radius of 10 Å). **c** The standard deviation of the full 3D-ΔPDF in shells of constant distance for each map slice shown in (**b**). Although GOODVIBES and experiment agree at large distances (beyond ~10 Å), a significant residual remains at shorter distances in all three datasets. The rapid decay of the residual component with inter-atomic distance is consistent among the three polymorphs.

Although far from perfect, these are some of the best correlations reported to date for a single unit-cell simulation[7]. Despite only being a single unit-cell simulation, MD is able to describe all aspects of the target map (mean, standard deviation, and precise pattern) remarkably well (Fig. 6b). Most importantly, we demonstrate a workflow for direct comparison between experimental diffuse scattering and a small-scale MD simulation. With this development, we can envision a way to experimentally validate atomistic simulations of protein motions.

## Discussion

Protein motions involved in key functions, such as catalysis, are thought to occur at the sub-angstrom to angstrom scale[21–25]. However, few experimental techniques are sensitive to the subtle intrinsic motions of proteins. Diffuse scattering from protein crystals has long been proposed as a promising approach as it contains information on the correlation of atomic displacements, i.e., how atoms move with respect to each other. Although the idea of using diffuse scattering was conceptualized decades ago[7,26], major bottlenecks stood in the way of realizing this as a general technique. With the recent introduction of direct X-ray detectors[12] and data collection and processing strategies[13], diffuse scattering data can now be measured with high accuracy. Before we can interpret the diffuse scattering signal from proteins, however, the more dominant signal arising from lattice disorder must be accounted for.

In this work, we developed a robust workflow for total scattering (TS) analysis that enables the complete characterization of correlated protein motion from RT-MX experiments. We demonstrated two applications with an immediate impact on X-ray crystallography. First, we are able to partition the ADPs into parts arising from internal and external motion (i.e. protein motion vs. lattice disorder). This enables a

more nuanced and unambiguous interpretation of structural heterogeneity, which is needed for making well-supported inferences about the role of dynamics in protein function. Likewise, we are able to partition the diffuse scattering into internal and external contributions. Consequently, correlated motion can now be compared between different crystalline environments, as we demonstrated with three lysozyme polymorphs, or between experiments and MD simulations of a single unit cell (Fig. 1).

It is remarkable how well the GOODVIBES lattice vibration model is able to reproduce experimental diffuse scattering features. Before this study, few examples of protein diffuse scattering of sufficient quality were available to test models of lattice disorder. Although the elastic vibrations model performed well in our previous work on triclinic lysozyme[13], the triclinic crystals have unusual properties, including a small unit cell, low solvent content, and low B-factors. However, tetragonal and orthorhombic lysozyme are more representative of a typical protein crystal according to statistics from the Protein Data Bank (PDB)[27]. Thus, the fact that elastic vibrations describe diffuse scattering from all three lysozyme polymorphs gives confidence that it can be applied generally. We note that there is nothing in our procedure that requires high-resolution Bragg data, however, the insights that can be derived from X-ray data depend on having high confidence in the average structure (e.g. as judged by R-factors). High resolution is already a precondition for detailed studies of biochemical mechanisms, and therefore we expect those experimental systems to be ideal candidates for TS/RT-MX.

When lattice disorder is adequately modeled, as judged using DISCOBALL, we have shown that its scattering contribution can be subtracted directly. The remaining signal from internal motion can then be interpreted using a model. Several parsimonious models have been proposed including elastic networks, normal modes, discrete

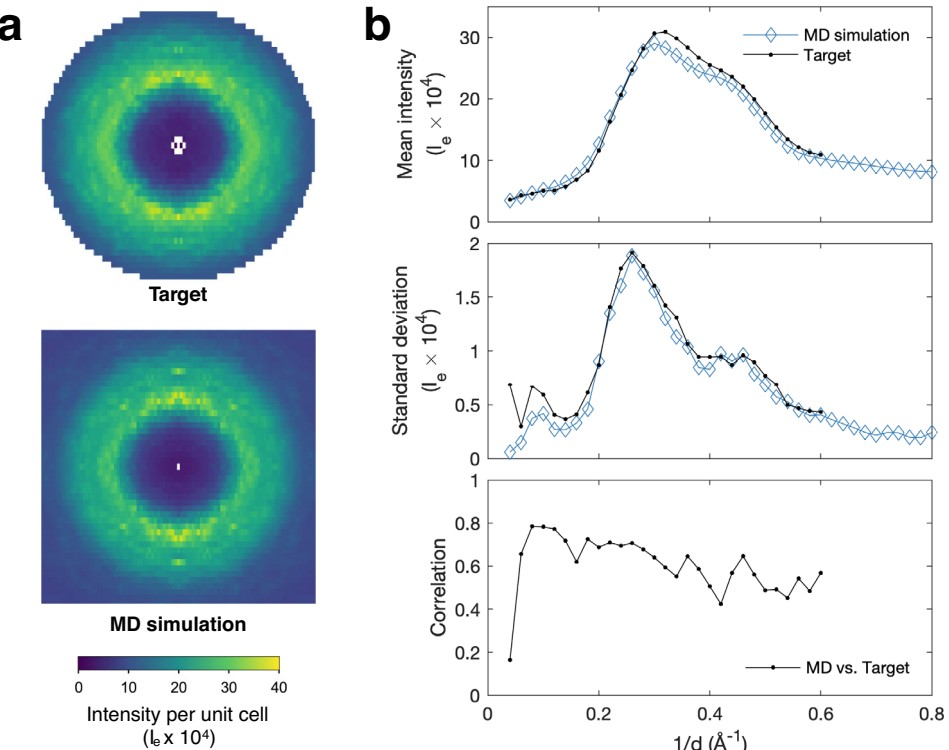

**Fig. 6 | Direct comparison between target diffuse map and crystalline MD.** MD simulations of tetragonal lysozyme were performed on a single unit cell with periodic boundary conditions (PBC) and a 3D diffuse map was simulated. **a** Slices through 3D maps of total diffuse intensity in the $k = 0$ plane from MD (bottom panel) and target map derived from experimental data and GOODVIBES fit (top panel) in electron units ($I_e$) per unit cell. See Methods and Supplementary Fig. 6a for details. **b** Direct comparison of MD and target diffuse maps in each resolution shell ($1/d$ is the scattering vector magnitude, $d$ is the resolution). The average diffuse intensity (top panel) agrees well between the target (black symbols) and MD (blue symbols). After subtracting the average (see Methods), the standard deviation of intensity is closely matched by MD (middle panel, black vs. blue symbols), including a distinctive shoulder at $d \sim 2$ Å. The Pearson correlation coefficient (CC, bottom panel) between MD simulation and target diffuse map is best at low resolution (CC ~ 0.8) and decays slightly at high resolution. Source data are provided as a Source Data file.

ensembles, and empirical correlation functions[7,8]. Until now, it has been difficult to evaluate these models experimentally because the signal from lattice disorder was present in the data[28,29]. Our TS/RT-MX workflow presents the possibility of directly refining a parsimonious model to the residual diffuse scattering. However, here we have chosen to highlight MD simulation as the method of choice going forward. It is the only method currently available that is able to simulate the entire diffuse signal; it excels at modeling hydration water[30] and disordered solvent[31] and it has the potential to explain biochemically relevant protein motions in detail[7]. MD also adds the time dimension, which is missing from our experiments but can be corroborated by spectroscopic probes such as solid-state NMR[32]. Here, we have laid the groundwork for greater synergy between MD simulation and total scattering by developing a procedure to produce target diffuse maps that can be compared directly with MD simulations of one or more unit cells with periodic boundary conditions.

Our application of MD simulation to the tetragonal lysozyme dataset demonstrates the potential for this approach. We found remarkable agreement on an absolute intensity scale for both the isotropic scattering component and the magnitude of scattering fluctuations, and reasonable agreement in the detailed fluctuation pattern (correlation coefficients vs. resolution of 0.5–0.8), including for features at 2 Å resolution attributed to internal motion. However, the discrepancies for diffuse maps between MD and experiment remain much larger than the experimental error. Thus, the accuracy of MD can potentially be improved using TS/RT-MX data as a target[5]. For instance, target diffuse maps could be used to derive external forces or

to bias the sampling by Monte-Carlo approaches. Our benchmark experimental datasets from lysozyme polymorphs will be instrumental in developing such methods.

Finally, by developing a robust TS/RT-MX workflow, we have laid the groundwork to study the role of correlated motion in enzyme function. In the context of protein allostery, correlated motions are implicated in propagating a signal from one site of a protein to another. Additionally, subtle motions in the active site of enzymes are thought to affect the rates of the reactions they catalyze. To examine these and other biochemical properties of proteins, high-resolution insight is needed. With the development of a general TS/RT-MX workflow, we can envision that this ultimate goal is within reach.

## Methods

### Crystallization, X-ray data collection, and structure refinement

Lyophilized hen egg white lysozyme from *Gallus gallus* (Sigma) was dissolved in 20 mM sodium acetate (NaOAc) pH 4.6 at a concentration of 100 mg/mL, passed through a 0.2 μm filter and subsequently diluted using the same buffer to the working concentrations (specified below). The tetragonal form was grown by hanging drop vapor diffusion (24-well VDX plate, Hampton Research). Drops containing 2 μL each of 40 mg/mL protein and reservoir solution were equilibrated over a reservoir of 1.2 mL at ambient temperature. Crystals were harvested from a well containing 1.1 M NaCl and 0.1 M NaOAc pH 4.8. The orthorhombic form was grown by sitting drop vapor diffusion (Cryschem plate, Hampton Research). Drops consisted of 10 μL of 90–100 mg/mL protein and 20 μL reservoir solution. The sealed trays

were incubated at 45 °C until crystals appeared, and then returned to ambient temperature. Crystals were harvested from drops equilibrated against a reservoir of 1.0–1.1 M NaCl and 0.1 M NaOAc pH 4.6.

Diffraction data were collected at the CHESS F1 beamline at ambient temperature[13]. Crystals were held in Kapton loops (MiTeGen Micro-RT) and surrounded by a PET capillary containing 10 μL of well solution in the tip to maintain hydration. The X-ray beam incident on the sample had a circular cross-section of 100 μm diameter, wavelength of 0.9768 Å, bandwidth $\Delta E/E \sim 5 \times 10^{-4}$, and flux of $\sim 2 \times 10^{11}$ photons per second. Diffraction images were recorded continuously using a pixel array detector (Pilatus3 6M, Dectris) at 10 frames per second while rotating the crystal about the phi axis at 1 degree per second (0.1 degree per frame). To avoid excessive radiation damage, the total exposure for each spot was limited to 50 seconds (50 degrees of oscillation over 500 frames). The tetragonal dataset consisted of 4000 frames acquired from 8 separate locations on a single large crystal. The orthorhombic dataset consisted of 4000 frames from 8 separate locations from two large crystals (4 locations on each). Background images were also collected for each crystal by translating the crystal out of the X-ray beam along the spindle axis, and acquiring images through the capillary at 1 frame per second while rotating at 1 degree per second (1 degree per frame).

Bragg peaks were indexed and integrated using *XDS*[33] and imported into *ccp4i2*[34] for further processing and structure refinement using the *ccp4* suite of programs[35]. Scaling and merging were performed using *Aimless*[36], and amplitudes were estimated using *ctruncate*. Initial phases were obtained by *Phaser*[37] using molecular replacement with search models from PDB IDs 193L (tetragonal) and 1WTM, and the structures were iteratively built and refined in *Coot*[38] and *refmac*[39]. Water, ions, and alternative conformers were added where supported by the electron density. Anisotropic ADPs were added in final rounds of refinement. Stereochemical quality was assessed using *MolProbity*[40] (Supplementary Table 2).

**Reconstruction of diffuse scattering maps from diffraction images**

The data reduction procedure was described previously for the triclinic lysozyme dataset[13]. Here we briefly describe the procedure and modifications for the tetragonal and orthorhombic datasets. Data reduction was performed in MATLAB using the macromolecular diffuse scattering library *mdx-lib*[13,41]. Each pixel on the detector was mapped to a fractional Miller index in reciprocal space using refined geometric parameters from *XDS* (*INTEGRATE.LP*). Photon counts were accumulated on a three-dimensional reciprocal space voxel grid. The datasets were processed in two passes as described below: first to generate a series of coarse maps for scaling, and second, to generate the fine, merged and scaled map of elastic scattering on an absolute scale (electron units per unit cell).

In the first pass, a coarse reciprocal space grid was used for computational efficiency. The grids were centered at the reciprocal lattice points (RLPs) and subdivided by an odd number of times along each reciprocal lattice direction. Odd numbered subdivisions were chosen so that each voxel is associated with an RLP (even subdivisions result in voxels mid-way between RLPs). The number of subdivisions in each direction was chosen to make the voxels approximately cubic (reciprocal cell dimensions are given in Supplementary Table 3). For tetragonal, a voxel size of (a*/3, b*/3, c*/5) was used for the coarse map. For orthorhombic, a voxel of (a*/5, b*/3, c*/3) was chosen. A count histogram was generated for each voxel, and a sensitive filter based on Poisson statistics was applied to detect and mask out non-smooth features, such as Bragg peaks. After masking, the grid was re-binned to one sample per RLP. The photon count rate was converted to intensity by applying geometric corrections for polarization, solid angle, detector quantum efficiency, and air absorption. Additionally, the voxels containing Bragg peaks were integrated, background

subtracted, and corrected for the geometric effects above plus the Lorentz effect.

A scaling model was refined to account for sample variations and artifacts[13]. The scaling model contained four terms: absorption, which depends on detector position and rotation angle; scale, which depends on rotation angle; detector chip gain, which applies globally to each of the 960 CMOS ASIC chips; and offset, which is a positive (excess) isotropic scattering correction that depends on rotation angle and scattering vector magnitude (or scattering angle). The continuous parameters were represented by linear interpolation of a regular grid of control points. Regularization was used to enforce smoothness and to minimize the contribution of the offset term, which would otherwise be underdetermined. The model parameters were refined to the diffuse scattering data to minimize the least-squares error between the observed and merged intensities of symmetry-related voxels, plus regularization terms weighted by Lagrange multipliers. The Lagrange multipliers were adjusted manually to obtain a physically reasonable and well-converged solution. The scaling models for orthorhombic and tetragonal lysozyme are shown in Supplementary Figs. 7 and 8. The scaling model was applied to both the Bragg and diffuse intensities, and each were merged separately. The Bragg intensities integrated using *mdx-lib* were then replaced by the more accurate values reported by *XDS/Aimless* after applying an appropriate scale factor.

The merged diffuse map includes Compton (inelastic) scattering in addition to the coherent (elastic) signal of interest. To place the data on an absolute intensity scale and subtract the Compton scattering, the integrated total scattering was compared with a theoretical value calculated from the unit cell's molecular inventory, which includes the disordered solvent. The number of waters and ions were estimated by equilibrating an MD simulation of a single unit cell, as described below, to achieve ~1 atm of pressure and to neutralize the charge. This resulted in 1890 water molecules and 32 Cl ions for orthorhombic, and 3304 water molecules and 56 Cl ions for tetragonal. For tetragonal lysozyme, the number of waters and ions differs slightly from the production simulation described below because of different assumptions about the lysozyme charge (the effect on scaling is negligible). The theoretically integrated intensity was computed for the model and data out to a maximum scattering vector of 0.7 Å⁻¹. The theoretical intensity included elastic scattering factors, incoherent scattering factors, and intramolecular interference terms due to covalent bonds as described previously[13]. A scale factor was found to bring the data into agreement with the model, and the Compton scattering was subtracted from the diffuse map.

Finally, in the second phase of data processing, we reconstructed maps of diffuse scattering on a fine grid by re-integrating the diffraction images and applying the scaling model determined from the first stage. The grids were subdivided an odd number of times per RLP, with subdivisions chosen to make the voxels approximately cubic, as described above for the coarse map, and the voxel volume comparable to the fine map reported previously for triclinic lysozyme (Supplementary Table 3). The diffraction images were re-integrated on the fine voxel grid, photon counts were corrected and then merged to the reciprocal space asymmetric unit according to the Laue group symmetry (Supplementary Table 3) using the geometric corrections and scaling model as described above. Voxels containing Bragg peaks were excluded. Finally, the data were placed on an absolute scale and the Compton scattering was subtracted. For visualization, the diffuse scattering was separated into isotropic and variational components as described previously[13]. Missing voxels in the variational scattering map (such as at Bragg peak locations) were filled using the mean value of the six nearest-neighbor voxels.

**GOODVIBES model**

Lattice disorder was modeled as thermally-excited vibrations of an elastic network, where each protein was described by a rigid body and

the inter-protein forces were modeled as harmonic springs. For each space group of lysozyme, a network of springs was generated from the atomic coordinates excluding hydrogen and solvent. A list of external contacts between each atom in the asymmetric unit and those of symmetry-related neighbors was computed using a KD tree-based search, where contact was defined using a hard distance cutoff of 4 Å. Symmetry operators were applied in order to expand this list to the entire unit cell. The contacts were modeled as generalized springs with a harmonic restoring force that depends on the relative displacement of the endpoints (see Model Parameterization and Refinement, below).

The motion of each rigid group (indexed by $\kappa$) in a particular unit cell (indexed by $l$) was described by a set of six generalized coordinates $\mathbf{w}_{\kappa l}$ that describe rotational and translational degrees of freedom. Rotations are defined with respect to a fixed origin for each rigid group $\mathbf{o}_\kappa$ (here, the center of mass of the rigid group). Let $\mathbf{r}_{j\kappa l}$ be the position of an atom $j$ in group $\kappa$ and unit cell $l$ relative to the unit cell's origin $\mathbf{R}_l$. The atom's instantaneous displacement is defined as $\mathbf{u}_{j\kappa l} = \mathbf{r}_{j\kappa l} - \bar{\mathbf{r}}_{j\kappa}$, where $\bar{\mathbf{r}}$ is the unperturbed position. The mapping from generalized coordinates to Cartesian displacements $\mathbf{u}$ can be approximated as a linear operator for small rotations (as in TLS refinement)[42] such that $\mathbf{u}_{j\kappa l} = \mathbf{A}_{j\kappa} \mathbf{w}_{\kappa l}$ where

$$\mathbf{A}_{j\kappa} \equiv \mathbf{A}(\bar{\mathbf{r}}_{j\kappa} - \mathbf{o}_\kappa) \qquad (1)$$

and

$$\mathbf{A}(\mathbf{x}) = \begin{pmatrix} 1 & 0 & 0 & 0 & x_3 & -x_2 \\ 0 & 1 & 0 & -x_3 & 0 & x_1 \\ 0 & 0 & 1 & x_2 & -x_1 & 0 \end{pmatrix} \qquad (2)$$

The thermally-excited vibrational motions of the crystalline elastic network were solved using the Born/Von-Karman (B/V-K) method[43,44] for a periodic supercell as described previously[13]. For notational convenience, the displacements of all $K$ groups in a particular unit cell are described by a $6K$-dimensional vector $\mathbf{w}_l$ formed by placing $\mathbf{w}_{\kappa l}$ end-to-end. The B/V-K method transforms the equations of motion into the form of an eigenvalue problem; the so-called dynamical matrix $\mathbf{D}_{(\mathbf{k})}$ is computed for each wavevector $\mathbf{k}$ and diagonalized to give $6K$ eigenvectors (normal modes) and corresponding eigenvalues (squared vibrational frequencies). The modes are then populated according to the classical (high temperature) equipartition theorem, resulting in a simple expression for the covariance of generalized coordinates:

$$\langle \mathbf{w}_l \mathbf{w}_{l'}^{\mathsf{T}} \rangle = N^{-1} k_B T \sum_{\mathbf{k}} \exp\left(i\mathbf{k} \cdot (\mathbf{R}_l - \mathbf{R}_{l'})\right) \mathbf{L}^{-\mathsf{T}} \mathbf{D}_{(\mathbf{k})}^{+} \mathbf{L}^{-1} \qquad (3)$$

where $N$ is the number of unit cells in the supercell, $\mathbf{k}$ runs over the allowed wavevectors in the first Brillouin zone, $k_B$ is the Boltzmann factor, $T$ is the temperature, $\mathbf{L}$ is a lower triangular matrix that depends on the masses and rotational moments of each group (defined previously[13]) and $\mathbf{D}_{(\mathbf{k})}^{+}$ is the pseudo-inverse of the dynamical matrix. The pseudo-inverse is identical to the inverse except when $\mathbf{k} = 0$: $\mathbf{D}_0$ is not invertible because it includes three modes with zero frequency corresponding to rigid translations of the entire supercell. In the pseudo-inverse, these three modes are removed by setting their reciprocal eigenvalues to zero.

The covariance matrix (Eq. 3) includes $6 \times 6$ diagonal blocks (where $l = l'$ and $\kappa = \kappa'$) composed of the $3 \times 3$ matrices $\mathbf{T}$, $\mathbf{L}$, and $\mathbf{S}$ (as defined in TLS refinement):[42]

$$\langle \mathbf{w}_\kappa \mathbf{w}_\kappa^{\mathsf{T}} \rangle = \begin{bmatrix} \mathbf{T}_\kappa & \mathbf{S}_\kappa^{\mathsf{T}} \\ \mathbf{S}_\kappa & \mathbf{L}_\kappa \end{bmatrix} \qquad (4)$$

where the $l$ index has been dropped because of translational symmetry. The atomic displacement parameters (ADPs) are found by

projection onto Cartesian coordinates:

$$\mathbf{U}_{j\kappa} = \mathbf{A}_{j\kappa} \langle \mathbf{w}_\kappa \mathbf{w}_\kappa^{\mathsf{T}} \rangle \mathbf{A}_{j\kappa}^{\mathsf{T}} \qquad (5)$$

Equivalent isotropic B-factors were computed from the ADPs as follows:

$$B_{\text{iso.}} = 8\pi^2 \text{trace}(\mathbf{U})/3 \qquad (6)$$

## GOODVIBES diffuse scattering simulation

Diffuse scattering from rigid-body motion was computed using the one-phonon approximation. In this approximation, which is valid for small displacements, lattice vibrational modes with wavevector $\mathbf{k}$ scatter independently and therefore contribute only at discrete points in reciprocal space (essentially, as satellite reflections at points $\mathbf{g}_\mathbf{h} \pm \mathbf{k}$, where $\mathbf{g}_\mathbf{h}$ points to an RLP with integer Miller indices $\mathbf{h} = (h,k,l)$). This enables rapid calculation of the diffuse halos when the Born/Von-Karman formalism is used to decompose the lattice dynamics in terms of vibrational modes, as described above. The one-phonon diffuse scattering, $I^{(1)}(\mathbf{q})$, where $\mathbf{q} = \mathbf{g}_\mathbf{h} - \mathbf{k}$, is computed as follows:

$$I^{(1)}(\mathbf{q} = \mathbf{g}_\mathbf{h} - \mathbf{k}) = k_B T\, \mathbf{G}(\mathbf{q})\left(\mathbf{L}^{-\mathsf{T}} \mathbf{D}_{(\mathbf{k})}^{+} \mathbf{L}^{-1}\right) \mathbf{G}^{\dagger}(\mathbf{q}) \qquad (7)$$

where $\mathbf{G}(\mathbf{q})$ is a vector-valued one-phonon structure factor with 6 components per rigid group. The components associated with a particular group $\kappa$ can be computed as follows:

$$\mathbf{G}_\kappa(\mathbf{q}) = \begin{bmatrix} \mathbf{q}\, \mathscr{F}^{-1}\{\bar{\rho}_\kappa(\mathbf{r})\}(\mathbf{q}) \\ -\mathbf{q} \times \mathscr{F}^{-1}\{(\mathbf{r} - \mathbf{o}_\kappa)\bar{\rho}_\kappa(\mathbf{r})\}(\mathbf{q}) \end{bmatrix} \qquad (8)$$

where $\bar{\rho}_\kappa(\mathbf{r})$ is the average electron density of group $\kappa$, $\mathbf{r}$ is the position relative to the unit cell origin, and $\mathscr{F}^{-1}$ is the inverse Fourier transform.

In ref. [13] the electron densities appearing in the one-phonon structure factor were computed from the refined atomic coordinates and ADPs. A potential disadvantage of this approach is that errors in the structural model propagate to errors in the simulated diffuse intensities. Here, we take a different approach, and compute the electron densities using model phases and experimentally determined amplitudes from the observed Bragg peak intensities. First, the unit cell electron density was computed by Fourier synthesis using model phases and experimental amplitudes. Then, the mean electron density in the bulk solvent region was subtracted from the map (by Babinet's principle, subtracting a constant does not alter the intensity except at $q = 0$). Next, the unit cell was partitioned into regions associated with each rigid group, creating a series of hard masks equal to 1 inside the rigid group and 0 outside. The hard masks were then blurred by convolving with a Gaussian function (B-factor of 50 Å$^2$) creating soft masks. The electron density of each rigid group was defined as the product of its soft mask and the unit cell electron density.

In order to compute the Fourier transforms efficiently for an arbitrary set of $\mathbf{q}$ points, an oversampling and Fourier interpolation scheme was used. The electron density of the rigid group ($\kappa$) was first shifted to the unit cell origin, and the grid was cropped or zero-padded to four times the maximum dimension of the group in each direction. Next, an inverse Fast Fourier Transform (IFFT) was performed to generate an oversampled molecular transform, $F_\kappa(\mathbf{q})$. The real and imaginary components of $F_\kappa(\mathbf{q})$ were then interpolated at the desired values of $\mathbf{q}$, and a phase factor was applied to undo the origin shift. A similar calculation was performed for the coordinate-weighted electron densities (one for each Cartesian coordinate). Finally, the components of $\mathbf{G}_\kappa$ were computed by taking the appropriate products between structure factors and Cartesian components of $\mathbf{q}$.

## GOODVIBES model parameterization and refinement

Elastic network models were optimized to fit the three-dimensional diffuse halos by allowing each of the spring constants to vary. Symmetry-equivalent springs were forced to have identical spring constants, and additional restraints were applied during refinement to prevent over-fitting. The functional form of the harmonic potential allows for different restoring forces for displacements perpendicular or parallel to the inter-atomic vector. Two special cases are defined: Gaussian, or equal restoring forces for any direction; and parallel, in which components of displacement perpendicular to the inter-atomic vector have zero restoring force. The general spring constant is parameterized using a linear combination of Gaussian and parallel forces (called hybrid springs).

A set of 400 halos was chosen from each experimental dataset corresponding to the most intense Bragg peaks in the 2.0 to 2.5 Å resolution range. The measured intensities on an absolute scale were extracted from the fine diffuse maps, along with experimental uncertainties propagated from photon-counting statistics. Spring constants were refined in four stages with progressively fewer restraints: (1) a single Gaussian spring constant was fit globally; (2) a Gaussian spring constant was fit for each unique protein-protein interface; (3) a hybrid spring constant was fit for each unique protein-protein interface; and (4) a hybrid spring constant was fit for each unique residue-residue interaction. The refinement program minimized the least-squares error between measured and simulated intensity using inverse-sigma weights derived from the experimental uncertainty. Since the one phonon intensity does not include the isotropic scattering, an additional linear background term was fit simultaneously for each halo (4 parameters per halo are added for the background term).

## Derivation of the 3D-ΔPDF peak function used by DISCOBALL

Here we derive an approximate function for the 3D-ΔPDF lattice peaks in the case of a crystal with space group symmetry. First we assume that the peaks arise mainly from correlated translational motions of rigid groups. We also assume that the displacement covariances are small so that the harmonic and one-phonon scattering approximations are valid. With these approximations, the diffuse scattering per unit cell is:

$$I_D(\mathbf{q}) = \sum_n e^{i\mathbf{q}\cdot\mathbf{R}_n} \sum_{\kappa,\kappa'} e^{i\mathbf{q}\cdot(\mathbf{r}_\kappa - \mathbf{r}_{\kappa'})} F_\kappa(\mathbf{q})F^*_{\kappa'}(\mathbf{q})(\mathbf{q}\cdot\mathbf{V}_{n\kappa\kappa'}\mathbf{q}), \qquad (9)$$

where the summation runs over unit cells in the crystal (labeled by $n$) and pairs of rigid groups (labeled by $\kappa$ and $\kappa'$), $\mathbf{R}_n$ is the unit cell origin, $F_\kappa(\mathbf{q})$ is the mean structure factor of a group with local origin at $\mathbf{r}_\kappa$, and $\mathbf{V}_{n\kappa\kappa'}$ is the (symmetrized) $3 \times 3$ covariance matrix for displacements of groups $\kappa$ and $\kappa'$ in unit cells separated by $\mathbf{R}_n$ (or *joint-ADP*). The 3D-ΔPDF is the Fourier transform of $I_D(\mathbf{q})$, as follows:

$$\mathcal{F}\{I_D(\mathbf{q})\}(\mathbf{r}) = \sum_n \sum_\kappa P_{n\kappa\kappa}(\mathbf{r} - \mathbf{R}_n) + \sum_n \sum_{\kappa,\kappa'\neq\kappa} P_{n\kappa\kappa'}(\mathbf{r} - \mathbf{R}_n - \mathbf{r}_\kappa + \mathbf{r}'_\kappa)$$
(10)

where

$$P_{n\kappa\kappa'}(\mathbf{r}) \equiv P_{\kappa\kappa'}(\mathbf{r}) * \mathcal{F}\{\mathbf{q}\cdot\mathbf{V}_{n\kappa\kappa'}\mathbf{q}\}(\mathbf{r}) \qquad (11)$$

and

$$P_{\kappa\kappa'}(\mathbf{r}) \equiv \mathcal{F}\{F_\kappa(\mathbf{q})F^*_{\kappa'}(\mathbf{q})\}(\mathbf{r}). \qquad (12)$$

The function $P_{\kappa\kappa'}(\mathbf{r})$ is the cross-correlation of mean electron density for pairs of groups in the unit cell, and $P_{\kappa\kappa'}(\mathbf{r})$ (when $\kappa = \kappa'$) is the autocorrelation. The autocorrelation has a strong peak at $\mathbf{r} = 0$, however the cross-correlations do not have strong peaks in general

because the groups are oriented differently in space (except in special cases such as translational non-crystallographic symmetry). Mathematically, the peaks at $\mathbf{R}_n$ are:

$$P_n(\mathbf{r}) = \sum_\kappa P_{\kappa\kappa}(\mathbf{r}) * \mathcal{F}\{\mathbf{q}\cdot\mathbf{V}_{n\kappa\kappa}\mathbf{q}\}(\mathbf{r}) \qquad (13)$$

When there is only one rigid body in the unit cell (P1 space group), $P_{\kappa\kappa}(\mathbf{r})$ is approximately equal to the Patterson function $P(\mathbf{r})$ for small $\mathbf{r}$, and thus we can write:

$$P_n(\mathbf{r}) \approx P(\mathbf{r}) * \mathcal{F}\{\mathbf{q}\cdot\mathbf{V}_n\mathbf{q}\}(\mathbf{r}) \qquad (14)$$

Thus, in the case of P1, each peak is simply a convolution of the Patterson function with another function that depends on the joint-ADP. Since the Patterson function is known from the Bragg data, deconvolution can be applied to recover the joint-ADPs without additional modeling.

In a symmetric space group, the deconvolution is more difficult because the contributions of each rigid group to the peak are mixed. We note that the 3D-ΔPDF likely contains sufficient information to perform this deconvolution if a sufficiently large region around each peak is modeled. However if too large a region is chosen, the contributions from the $\kappa\neq\kappa'$ terms also become significant. Thus, here we restrict our fits to a limited region around the peak (small $\mathbf{r}$).

The $P_{\kappa\kappa}(\mathbf{r})$ are identical except for a rotation operator, and at small $\mathbf{r}$, they contain a strong peak that is dominated by individual atom contributions (Supplementary Fig. 9a, left panel). This peak region is expected to have approximate rotational symmetry except in special cases (such as when the resolution is highly anisotropic). Thus, in the general case, the peak does not contain enough information to deconvolve the contributions of each rigid group individually. Instead, we assume that the peak can be approximated by its symmetry average, or equivalently the Patterson peak for the unit cell (Supplementary Fig. 9a, middle and right panels). Then, the peak shape is described by Eq. (14) with an effective joint-ADP that is equal to the average over groups, as follows:

$$V_n^{(\text{effective})} = \frac{1}{K}\sum_{\kappa=1}^K V_{n\kappa\kappa}. \qquad (15)$$

Modeling the symmetric crystal as though it were P1 involves assumptions that are not obviously valid. However, lattice dynamics simulations show that the error incurred is small enough to be neglected in the cases studied here (Supplementary Fig. 9b).

Finally, we note that the DISCOBALL formula (Eq. (14)) resembles the liquid-like model (LLM) for diffuse scattering[7,26,45]. The LLM is derived by assuming that joint ADPs depend only on the inter-atomic vector. We do not make that assumption explicitly when deriving DISCOBALL, however doing so has certain advantages; it would ensure that Eq. (14) applies regardless of space group symmetry, and that Eq. (14) remains valid for the entire 3D-ΔPDF, not just the peak regions. Our derivation, though more complex, uses a more limited set of assumptions so that joint ADPs are directly comparable with those derived from atomistic models such as GOODVIBES.

## DISCOBALL deconvolution method

Given a 3D-ΔPDF and Patterson function, we can fit the effective joint-ADPs as follows. First, crop out a spherical region (radius $d$) of the 3D-ΔPDF centered on $\mathbf{R}_n$ and identify this region with $P_n(\mathbf{r})$:

$$\mathbf{1}_{|\mathbf{r}|<d}P_n(\mathbf{r}) = \mathbf{1}_{|\mathbf{r}|<d}P(\mathbf{r}) * \mathcal{F}\{\mathbf{q}\cdot\mathbf{V}_n\mathbf{q}\}(\mathbf{r}) \qquad (16)$$

where $\mathbf{1}_{|\mathbf{r}|<d}$ is an indicator function representing a spherical mask with radius $d$. By applying the inverse Fourier transform to both sides of this

equation, we can transform the deconvolution problem into ordinary least squares. Let

$$Y_n = \mathcal{F}^{-1}\{\mathbf{1}_{|\mathbf{r}|<d}P_n(\mathbf{r})\}(\mathbf{q}) \tag{17}$$

and

$$X = \mathcal{F}^{-1}\{\mathbf{1}_{|\mathbf{r}|<d}P(\mathbf{r})\}(\mathbf{q}) \tag{18}$$

be the inverse Fourier transforms of the masked peaks and density autocorrelation, respectively. The six unique components of the joint-ADP can then be found by least squares minimization:

$$\hat{\mathbf{V}}_{\mathbf{n}} = \underset{\mathbf{V_n}}{\arg\min} \int_D d\mathbf{q}|Y_n(\mathbf{q}) - X(\mathbf{q}) \cdot [\mathbf{q} \cdot \mathbf{V}_n\mathbf{q}]|^2 \tag{19}$$

The domain of integration $D$ depends on the resolution of the data. In our implementation, $P(\mathbf{r})$ and $P_n(\mathbf{r})$ are represented on a discrete real-space grid, and thus the integral becomes a sum over a set of samples in reciprocal space.

## DISCOBALL analysis of experimental datasets
To calculate each 3D-ΔPDF, the variational component of the measured diffuse scattering was truncated to a resolution of 1.6 Å, missing values were filled with zeros, and the 3D-ΔPDF was calculated using a fast Fourier transform (FFT). Peaks regions were extracted from the 3D-ΔPDF and a spherical mask was applied to each with a maximum radius of $d = 4$ Å. The regular Patterson map was computed by taking the FFT of the Bragg intensities. The central peak was interpolated onto the same real space grid as the 3D-ΔPDF peaks and the same mask was applied.

For each 3D-ΔPDF peak, the 6 unique values of $\mathbf{V_n}$ were found by least squares fitting. We observed that the very low resolution points deviated from the expected behavior, likely because the constant offsets in the 3D-ΔPDF or Patterson map are not exactly known. Thus, points with d-spacing greater than 5 Å were excluded from the fit. Minimization was performed for each peak using the *lsqnonlin* function in MATLAB.

## DISCOBALL validation of GOODVIBES models
For each GOODVIBES simulation, the center-of-mass covariances were calculated for each pair of rigid bodies as follows:

$$\mathbf{U}_{\kappa l\kappa' l'} = \mathbf{A}_\kappa(\mathbf{0})\langle\mathbf{w}_{\kappa l}\mathbf{w}_{\kappa' l'}^\mathsf{T}\rangle\mathbf{A}_{\kappa'}^\mathsf{T}(\mathbf{0}) \tag{20}$$

$\mathbf{A}_\kappa(\mathbf{0})$ projects from the generalized coordinates onto the Cartesian displacement at the center of mass of rigid body $\kappa$ (see Eq. 1). The joint-ADP for a pair of rigid bodies is defined as the symmetrized covariance matrix:

$$\mathbf{V}_{n\kappa\kappa'} = \frac{1}{2}\left(\mathbf{U}_{\kappa l\kappa' l'} + \mathbf{U}_{\kappa' l'\kappa l}\right) \tag{21}$$

where the unit cell index $n$ refers to the relative displacement of unit cells $l$ and $l'$, $\mathbf{R}_n = \mathbf{R}_{l'} - \mathbf{R}_l$.

For cross-validation using DISCOBALL, the effective joint-ADPs were computed from the simulation using Eq. (15). To compare joint-ADPs between DISCOBALL and model calculations, Pearson correlation coefficients were calculated for isotropic and anisotropic components of the joint-ADPs. The isotropic part (or total covariance) was defined as

$$V_{\mathbf{n}}^{\text{total}} = \text{trace}(\mathbf{V_n}) \tag{22}$$

and the anisotropic (residual) components were

$$\mathbf{V}_{\mathbf{n}}^{\text{aniso}} = \mathbf{V_n} - \mathbf{I}_3 V_{\mathbf{n}}^{\text{total}}/3 \tag{23}$$

where $\mathbf{I}_3$ is a $3 \times 3$ identity matrix.

## MD simulations
A simulation of a single unit cell (8 protein chains) of tetragonal lysozyme was performed, in a manner similar to the triclinic lysozyme simulations described previously[13,15]. We used *Amber18*, with the ff14SB force field for the protein[46,47] and the OPC model for water[48]. The starting structure was taken from PDB entry 5L9J[49], using the *A* alternate conformer. We added 32 chloride ions to neutralize the charge, and 2540 additional water molecules to fill in space. The number of water molecules was manually adjusted in order to achieve -1 atm pressure at 295 K, resulting in 3288 total waters (411 per protein chain). The resulting density was 1.23 g/cm³, which is the same as a measured density for crystals grown from an NaCl soak density of 1.05 g/cm³[50].

The simulations were equilibrated under NVT conditions for 0.6 μs and continued for an additional 5 μs, saving coordinates every 0.4 ns. A time step of 4 fs was used, where non-water hydrogen masses are set to 3 amu, with a corresponding decrease in the mass of its bonded atom[51]. Diffuse scattering was computed as described earlier[13].

Since X-rays scatter from individual electrons, it is better to account for conformational disorder and flexibilty by averaging the electron density rather than the structures themselves. We took the average structure factors from the simulation (truncated to 1.5 Å, which is the experimental resolution limit for 5L9J), and reset the symmetry to P4₃2₁2. Using the deposited 5L9J pdb model as a starting structure, we ran 15 cycles of phenix.refine with default parameters. The backbone RMS deviation between this structure and the 5L9J crystallographic model was 0.44 Å.

## Procedure to create target diffuse maps for comparison with MD
A target map matching the one unit cell MD of tetragonal lysozyme was created from the experimental data in several processing steps (Supplementary Fig. 6a). A map of scattering from internal motion was found by subtracting the GOODVIBES supercell scattering simulation from the experimental diffuse map and then applying a filter. This filter served two purposes: first to re-sample the map on an appropriate reciprocal space grid for comparison with MD, and second to mitigate the effect of noise and outliers.

We implemented a robust, error-weighted Savitsky-Golay filter[17] as follows. First, a target grid was chosen with sufficient sampling to describe the molecular transform. For tetragonal lysozyme, a target grid with voxel dimensions of (a*, b*, c*/2) was used. Then, for each voxel in the target grid, a spherical neighborhood of voxels in the parent grid was defined using a cutoff radius of $(3\upsilon_t/(4\pi))^{1/3}$ where $\upsilon_t$ is the volume of the target voxel. For each neighborhood, a second-order polynomial was fit using weighted least-squares[17]. Then, the value of the polynomial was computed at the target grid location, and the corresponding uncertainty was propagated from the experimental errors. To render the method robust to outliers, the fit was performed a second time with adjusted weights. The re-weighting factors were chosen using the bisquare method[52] which is based on residuals from the first pass. To compute the residual at each point in the parent grid, we first performed cubic interpolation of the intensities from the target grid back to the original grid. Then, the bisquare re-weighting factor was computed for each parent voxel ($i$) as follows[52]:

$$w_i = \begin{cases} \left(1 - \left(\frac{r_i}{6m}\right)^2\right)^2 & |r_i| < 6m \\ 0 & \text{otherwise} \end{cases} \tag{24}$$

where $m$ is the median absolute deviation of the inverse sigma-weighted residuals $r_i$. Finally, the bisquare weights were applied and fits were repeated, yielding the filtered internal map on a coarse grid.

After obtaining the internal motion map, scattering from external motion compatible with the MD supercell was simulated and added back. To match the single unit cell MD simulation of tetragonal lysozyme, the reciprocal lattice points with integer Miller indices were selected from the internal map. Next, the scattering from GOODVIBES simulation corresponding to the MD supercell was added. For a one unit cell simulation, these are the k = 0 modes (optical phonons in which protein molecules move out of phase with each other such that the center of mass is stationary). Note that a new GOODVIBES simulation was not required in this case, because scattering from k = 0 modes was included in the previous supercell simulation (at integer Miller index points).

Finally, to compare MD simulation and target diffuse maps, the mean intensity, standard deviation (SD), and correlation coefficient (CC) were computed within resolution bins of width 0.02 Å$^{-1}$. Since the isotropic scattering changes significantly across certain bins, it potentially contributes to the SD and CC if not subtracted first[20]. Therefore, for each diffuse map, we fit a smooth one-dimensional interpolant to the intensity vs. resolution, and subtracted this isotropic signal before computing SD and CC.

### Reporting summary
Further information on research design is available in the Nature Portfolio Reporting Summary linked to this article.

### Data availability
The raw diffraction data generated in this study have been deposited in the SBGrid Data Bank under accession codes 957 (tetragonal lysozyme)[53] and 958 (orthorhombic lysozyme)[54]. The atomic coordinates and structure factors have been deposited in the Protein Data Bank (PDB) with accession codes 8dyz (tetragonal lysozyme) and 8dz7 (orthorhombic lysozyme). For the previously published triclinic lysozyme dataset[13], raw diffraction images are available from the SBGrid Data Bank under accession code 747[18], processed maps are available from the Coherent X-ray Imaging Data Bank (CXIDB) under accession code 128[19], and atomic coordinates and structure factors are available from the PDB with accession codes 6o2h. The source data underlying Fig. 6 are provided as a Source Data file. Starting coordinates for molecular replacement and MD simulations were obtained from the PDB entries and 193l, 1wtm, and 5l9j. Source data are provided with this paper.

### Code availability
Structure determination was performed using software curated by SBGrid[55]. Subsequent data reduction, model fitting, and analysis was performed using MATLAB with *mdx-lib*. The software is available on GitHub at https://github.com/ando-lab/mdx-lib and version 1.2.0 used in this study has been deposited in Zenodo[41]. Scripts required to reproduce the analysis using MATLAB are available in a separate GitHub repository https://github.com/ando-lab/mdx-examples, and version 0.1.1 used in this study has been deposited in Zenodo[56]. The script md2diffuse.sh used to compute diffuse scattering from MD trajectories is distributed with AmberTools (http://ambermd.org).

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

## Acknowledgements

The authors thank Syed Muhammad Saad Imran for growing the crystals that yielded the orthorhombic data used in this study. We are grateful to Dr. Veronica Pilar and Dr. Will Thomas for assistance with data collection, and Dr. Xiaokun Pei, Haoyue Wang, and Neti Bhatt for critical feedback on the manuscript. We are especially grateful to Mike Wall for his feedback on our preprint and for suggesting an improved method for calculating correlation coefficients with MD. Experiments were performed at the Center for High Energy X-ray Sciences (CHEXS), which is supported by the National Science Foundation (BIO, ENG and MPS Directorates) under award DMR-1829070, and the Macromolecular Diffraction at CHESS (MacCHESS) facility, which is supported by award 1-P30-GM124166-01A1 from the National Institute of General Medical Sciences, National Institutes of Health (NIH), and by New York State's Empire State Development Corporation (NYSTAR). This work was supported by NIH grants F32GM117757 (to S.P.M.), GM122086 (to D.A.C.), and GM124847 (to N.A.) and startup funds from Cornell University (to N.A.).

## Author contributions

S.P.M. performed the experiments, developed the GOODVIBES and DISCOBALL software, and analyzed the data. D.A.C. performed and analyzed the MD simulations. N.A. conceived of and supervised the research. All authors contributed to writing the paper.

## Competing interests

The authors declare no competing interests.
