## [Peer Review File · Nature Communications]

Robust total X-ray scattering workflow to study correlated motion of proteins in crystalsReviewers' Comments:

Reviewer #1:

Remarks to the Author:

See attached .pdf

Overview

The authors have used both an elastic network model and a phenomenological model to describe diffuse haloes in experimental X-ray diffraction data from lysozyme crystals grown in three space groups. The diffraction data were processed using `mdx-lib` software to yield 3D total scattering datasets with either coarse (for comparison to MD) or fine (for development of lattice dynamics models) sampling of reciprocal space. The spring constants of an elastic network model (GOODVIBES) were refined to optimize the agreement with 400 haloes selected from each finely sampled dataset. The phenomenological model (DISCOBALL) was developed by extracting the diffuse component of the total scattering, computing the Fourier-transform to yield the 3D- Δ PDF, and fitting a protein displacement covariance matrix (joint ADPs) associated with different lattice vectors, using a deconvolution method. The authors demonstrate that the covariance matrices obtained from GOODVIBES and DISCOBALL are consistent with each other. They use their approach to interpret MD simulations of diffuse scattering from tetragonal lysozyme crystals, and, using correlation coefficients as a metric, show that the simulations include contributions from both lattice vibrations and internal motions.

Specific comments

- p. 12, line 265. "...these are some of the best correlations reported to date for a single unit-cell simulation." The "subtracted experiment" comparison in the bottom panel of Fig. 6b with correlations ~ 0.4 is on par with previous MD simulations using a single unit cell, and is lower than what has been reported previously using multiple unit cells ($> \sim 0.6$). To create the "corrected experiment" where the correlations are higher, the authors have filtered out the haloes from the data, and then added back in the simulated haloes from the GOODVIBES model. The result is a combination of model and data and not "corrected" in the usual sense (e.g. calibrated or transformed to decrease a systematic error in the signal). One could just as well say the data was filtered to remove the haloes, and combined with the GOODVIBES model to create an enhanced model. The "corrected experiment" comparison therefore really isn't really a comparison with experimental data and should be renamed ("data-enhanced GOODVIBES model"?) to avoid confusion.
- p. 12, line 267-268. "In our case, the strong agreement stems not from an improved force field per se, but from a more sophisticated data treatment." The correlation of the MD with the "corrected experiment" is comparable to the "rigid body modes" below about 3 Å resolution, and both are substantially higher than the "subtracted experiment" in this range. At higher resolution, the correlations of the MD with the "subtracted experiment" and "rigid body modes" are comparable, and the correlation with the "corrected experiment" is increased compared to either alone. This means that most of the strong agreement with the "corrected experiment" comes from the GOODVIBES model and not from a more sophisticated data treatment (see above, the "corrected data" is not really the result of a data treatment, as it involves adding a model to the data).
- p. 13, Fig. 6. The labels in the figure legends are a little confusing. According to the caption, the green symbols correspond to the filtered diffuse data (processed to remove the haloes). However, in the panel legend it is described as "Subtracted experiment" or "subtracted experimental map". The blue corresponds to the GOODVIBES model fit to the haloes, but the caption just describes it as "Rigid body modes," which doesn't completely identify it as a model. See also previous comment about renaming "corrected experiment".
- p. 13, Fig. 6b, top panel. The agreement of the MD simulation with the isotropic data is remarkable. It is not yet perfect, however. The agreement of the isotropic component of the data with the MD is best at high res (above the peak) for the "subtracted experimental map", and best at low res for the "corrected experiment". Do the authors have any insights into why adding in the GOODVIBES model should increase the agreement at low resolution but decrease the agreement at high resolution? It would be nice to highlight this result a bit more in the manuscript, it's impressive, even though there's still room for improvement.
- p. 21, line 511. It is not obvious that the cross terms between groups can be neglected. A more natural way of proceeding here would be to assume that the $\mathbf{V}_{nkk'}$ are the same for all pairs of groups, so that $\mathbf{V}_{nkk'} \rightarrow \mathbf{V}_n$. With $\mathbf{V}_{nkk'} \rightarrow \mathbf{V}_n$ Eq. (9) becomes

$$I_D(\mathbf{q}) = N \sum_n e^{i\mathbf{q}\cdot\mathbf{R}_n} (\mathbf{q}\cdot\mathbf{V}_n\cdot\mathbf{q}) \sum_{\kappa,\kappa'} e^{i\mathbf{q}\cdot(\mathbf{r}_\kappa-\mathbf{r}_{\kappa'})} F_\kappa(\mathbf{q}) F_{\kappa'}^*(\mathbf{q}).$$

Note: The factor of N is missing in the authors' equation, I believe this is needed to account for the reduction of the double sum to a single sum over lattice points. The Fourier transform of this equation immediately yields Eq. (14) in the manuscript for all space groups (again, up to a factor N), without further assumptions:

$$\begin{aligned} \text{3D-}\Delta\text{PDF}(\mathbf{r}) &= N \sum_n \mathcal{F}\{\mathbf{q}\cdot\mathbf{V}_n\cdot\mathbf{q}\} * \delta(\mathbf{r}-\mathbf{R}_n) * P(\mathbf{r}) \\ &= N \sum_n \mathcal{F}\{\mathbf{q}\cdot\mathbf{V}_n\cdot\mathbf{q}\} * P(\mathbf{r}-\mathbf{R}_n) \end{aligned}$$

Here, $P(\mathbf{r})$ is the unit cell Patterson. Changing the math in this way would simplify the presentation, and connects the DISCOBALL model with the liquid-like motions model, with which it is closely related. The DISCOBALL model is distinguished from the liquid-like motions model in that it assumes that all pairs of atoms between the same two unit cells have the same covariance matrix of displacements (joint ADPs), whereas the liquid-like motions model assumes that the covariance matrix \mathbf{V} is a continuously varying function of the separation between atoms in the crystal.

- p. 21, last para. It's not clear what Patterson is used for the calculations in symmetric space groups. Is it the Patterson of the P1 unit cell, or is it a symmetrized Patterson computed just from the asymmetric unit? With the assumption $\mathbf{V}_{nkk'} \rightarrow \mathbf{V}_n$ mentioned above, the Patterson of the P1 unit cell would be well justified.
- p. 22, Eq. 15 – Each pair of equivalent groups in different unit cells can be superimposed on each other equivalent group via a translation and rotation of the crystal. Assuming any differences in the morphology of the crystal in different orientations can be neglected, all of the covariances between equivalent groups in different unit cells should, in principle, be the same.
- p. 22, line 523. The different $P_{kk}(\mathbf{r})$ will only be similar up to a symmetry transformation.

Summary

The major innovation of this paper is the DISCOBALL method, which enables lattice displacement covariance matrices (joint ADPs) \mathbf{V}_n to be extracted in a data-driven way from the computed 3D- Δ PDF. Because they are able to fit multiple such matrices using the data, the amount of information the authors extract from the data using this method far exceeds that previously obtained using liquid-like motions (LLM) models, which assume a simple functional dependence of the covariance matrix on the separation between atoms. For example, Clarage et al (1992) used four parameters to describe isotropic haloes plus cloudy diffuse features in lysozyme, and Wall et al (1997) used 6 parameters (the model contained 12 but the principal axes were fixed, eliminating 6) to describe the diffuse streaks in calmodulin. Here the authors express each joint ADP using 6 variables; from the scatter plots in Fig. 4 there appear to be $O(10^2)$ joint ADPs that were fit (I cannot find the numbers, it would be good to provide these somewhere if not already in the ms), meaning that there might be more than 10^3 joint ADP variables that come out of the DISCOBALL fits for each dataset. This information provides a detailed picture of the lattice dynamics and should be useful both for gaining insight into the internal dynamics of the protein and for explaining all of the X-rays in the diffraction experiment, which have long been goals of protein diffuse scattering research.

The part of the manuscript that deals with the use of GOODVIBES to compare the MD simulations to the data is an interesting addition but needs some revision. I do believe the methods described will be able to help to extract information about internal motions, but right now it includes a comparison between the MD and a data-enhanced *model* that is instead being represented as a comparison between the MD and the *data*, along with a claim that the agreement is among the best reported previously for similar simulations. I understand what the authors are trying to do here, but I think this comparison needs to be presented in a different way to remove any confusion and to avoid misleading the reader. There might also be other comparisons that would demonstrate the ability to extract information

about internal dynamics. For example, might it be possible to remove the lattice dynamics from the MD and then compare to the subtracted experiment? Or if the authors can somehow combine just the internal motions of the MD with their GOODVIBES model and compare to the total diffuse signal, that might also prove to be a good approach. There are multiple ways of enhancing the presentation in this section — the most important thing is to increase the clarity and decrease confusion. There are currently only a couple of paragraphs in the results dedicated to the MD, and although it would be interesting to learn more here, it's not really necessary; the DISCOBALL/GOODVIBES results are very important and can carry the paper.

Michael Wall

Reviewer #2:
Remarks to the Author:

The authors describe a significant breakthrough in the development of their X-ray crystal diffuse scatter analysis software. A key hurdle to this type of analysis is separating contributions from the "interesting" motions, such as functionally-relevant correlated movements within the molecule to the "uninteresting" diffuse scatter that comes from defects and other disorder in the crystal lattice itself. The authors make this distinction by declaring "lattice disorder" to be rigid-body movements of the molecules in the crystal. In this case there is only one peptide chain per enzyme, so the distinction is clear. Focusing on the halos that surround the Bragg spots, they fit the spring parameters in an elastic network model to a subset of these halo shapes with their "GOODVIBES" program, achieving an excellent match to all halos for not just one but three different crystal forms of lysozyme. This model then allows the "lattice disorder" component of the diffuse scatter to be simulated and then subtracted off the observed pattern, leaving behind the weaker but more "interesting" (to enzymologists) short-range correlated motion signal.

They also directly transform the halo data using a different approach called "DISCOBALL". This may not be quite as independent as the authors posit, since it is the same halos being either fit or transformed, but the algorithm is indeed radically different, lending confidence to the correctness of both implementations.

The impact of this achievement is potentially very exciting. Many enzymologists and others biologists with an interest in function will want to try this on their favorite systems. Although there are other diffuse scattering software packages that have been around for a while, these authors are neo-pioneers in the field, and have now solved many of the problems of data collection and data reduction for diffuse scatter X-rays. Here they demonstrate convincingly a key advance in data analysis, which is the separation of long- and short- range correlated motion. Even if the halos are being "over fit" by the admittedly complex underlying mathematical structure, this is still a good way to separate these two types of movement. I predict this will be a very high-impact paper.

A few specific comments.

Use of the word "subtle" to describe the correlated movements that make enzyme function possible might be inappropriately underwhelming. Perhaps a stronger word such as "key" or "critical" or "functional" would be better?

"not yet been realized." - this is not the first diffuse-scatter software package. might be politic to mention the closest anyone has gotten? I recommend generally against the use of "priority language" as it is difficult to back up and ultimately has little archival value.

"accounted for entirely by physical models,". - confusing - is it "entirely accounted" or "entirely physical" ? If the former, I don't think even the present work is accounting for everything, and if the latter, are previous formalisms like the "liquid-like-motions" really non-physical? This sentence needs to be better.

"covariances for pairs of protein chains" - not atoms? Does not the covariance change from atom to atom? This threw me.

"absolute intensity" - please provide units at first mention

"maximum tolerable dose was set conservatively" - please provide the actual dose in Gy (J/kg)

Fig 3: "central sections (middle vs. bottom row)" Which is which? real vs sim? Suggest adding labels to figure

"corrected map" - unclear - although described more thoroughly in the extended section, the

description in the main text is ... hard to follow. This reviewer had to read it many times before really understanding what the "corrected map" is. And even then: what filter? In cases such as this a statement as to the motivation for the "corrected map" can be helpful. Such as "we added back the lattice-disorder DS to the MD-derived DS for a more direct comparison to the experimentally observed DS". Or something along those lines.

"hen egg white" - ambiguous - replace with species name. I.E. an adult female *Meleagris gallopavo* is called a "hen", but I suspect the protein used here was from *Gallus gallus* ?

experimental amplitudes - why not use 2mFobs-DFcalc map? This map is the closest to the "true" electron density. Much closer than an Fo map phased with PHCalc. Although it is a clever advance to use Fobs, I predict that using the standard sigma-A weighted maps will be better. Might need to be scaled to be on the same scale as Fcalc, but otherwise should be compatible with the algorithm as described.

Note: I do not present this as a requirement for publication, but merely a suggestion to the authors. Should they go ahead with the Fobs@PHCalc density, however, perhaps a sentence as to why the 2mFobs-DFcalc map was not used would be in order.

"averaging the averaging the electron density" - duplicate phrase

"the value of the intensity and its uncertainty were computed at the target grid location"
would it be more accurate to say the intensity and uncertainty were interpolated to the target grid? Or perhaps I misunderstand?

Extinction? or pile-up? The non-linear disagreement between Fobs and Fcalc noted in the extended figures could be due to something other than extinction. With crystals of this size, extinction would be surprising. An alternative and testable hypothesis is that the low mosaic spread of these room-temperature crystals is resulting in very narrow rocking widths that overwhelm the detector's maximum count rate. For the Pilatus3 used here this is $1e7$ photons/s/pixel. The "seconds" used here are not the exposure time, but rather the time it takes the relp to transit the Ewald sphere, which can be very short for narrow mosaic crystals in low-divergence beams. Since the authors are working on the absolute photon scale already, it should be a simple matter to check how close the instantaneous maximum count rate came to the detector's rated maximum. Yes, Pilatus3 has the "instant retrigger" feature, but residual non-linearity can still happen despite this correction. Alternately, the observation of this non-linearity does not contribute to the main thrust of this paper, and it would be equally impactful (and shorter) if it were left out.

Well done all, and I look forward to the final version in print!

Reviewer #3:

Remarks to the Author:

Review Robust total X-ray scattering workflow to study correlated motion of proteins in crystals

Short

The paper introduces methodology for fitting diffuse halos around Bragg peaks of protein crystals. It addresses an important problem of using single crystal diffuse scattering for understanding the protein dynamics. Diffuse scattering is a continuous signal which is always present alongside Bragg peaks, and presents information about variability in the structure. In the case of proteins, diffuse scattering probes contain confirmations that are dynamically accessible for the crystallized protein which in many cases is related to the protein function. Unfortunately, due to complexity of modeling diffuse

scattering, information which it contains is currently ignored.

The paper presents an important progress to solving this problem. It presents two methods for fitting a portion of diffuse scattering which is present in a form of halos around Bragg peaks. The methodology is general and applicable to most of the proteins. It is implemented in well-written and well-documented scripts and in my view has a great potential to be widely adopted in analysis of protein diffuse scattering. I believe that the presented methods will have a major impact, and I recommend publishing the paper with minor corrections.

Minor comments

Page 10, figure 4a: Would it be possible to add a faint gray line at the covariance = 0 \AA^2 in order to visually highlight correlations approaching negative values for long pairs? Please note that in my view the negative values of the correlations are completely justified and their origin is well explained in the text (page 7 198-211).

Page 31, Extended figure 4: Colormap is inverted (brown-negative, purple - positive) wrt colormap 5b. Could you flip the colors in the figure 4?

Also what are the units that you use in figure 4? Shouldn't autocorrelation be expressed in electrons squared per volume (e^2/vol). Please note that in my view it is not important to find the absolute scale, relative scale (say $P/\max(P)$ where P is the 3D- Δ PDF is sufficient in this case.

:

Please find enclosed our revised manuscript, “Robust total X-ray scattering workflow to study correlated motion of proteins in crystals.” All of the reviewers’ comments are reproduced verbatim and addressed point-by-point below. For clarity, reviewer comments are in black and our response is in blue. Revisions to the manuscript are highlighted in yellow.

Reviewer 1

Overview

The authors have used both an elastic network model and a phenomenological model to describe diffuse haloes in experimental X-ray diffraction data from lysozyme crystals grown in three space groups. The diffraction data were processed using mdx-lib software to yield 3D total scattering datasets with either coarse (for comparison to MD) or fine (for development of lattice dynamics models) sampling of reciprocal space. The spring constants of an elastic network model (GOODVIBES) were refined to optimize the agreement with 400 haloes selected from each finely sampled dataset. The phenomenological model (DISCOBALL) was developed by extracting the diffuse component of the total scattering, computing the Fourier-transform to yield the 3D- Δ PDF, and fitting a protein displacement covariance matrix (joint ADPs) associated with different lattice vectors, using a deconvolution method. The authors demonstrate that the covariance matrices obtained from GOODVIBES and DISCOBALL are consistent with each other. They use their approach to interpret MD simulations of diffuse scattering from tetragonal lysozyme crystals, and, using correlation coefficients as a metric, show that the simulations include contributions from both lattice vibrations and internal motions.

Specific comments

- [point 1] p. 12, line 265. “...these are some of the best correlations reported to date for a single unit- cell simulation.” The “subtracted experiment” comparison in the bottom panel of Fig. 6b with correlations ~ 0.4 is on par with previous MD simulations using a single unit cell, and is lower than what has been reported previously using multiple unit cells ($> \sim 0.6$). To create the “corrected experiment” where the correlations are higher, the authors have filtered out the haloes from the data, and then added back in the simulated haloes from the GOODVIBES model. The result is a combination of model and data and not “corrected” in the usual sense (e.g. calibrated or transformed to decrease a systematic error in the signal). One could just as well say the data was filtered to remove the haloes, and combined with the GOODVIBES mode to create an enhanced model. The “corrected experiment” comparison therefore really isn’t really a comparison with experimental data and should be renamed (“data-enhanced GOODVIBES model”?) to avoid confusion.

We thank the reviewer for raising this point. Our description of how the “corrected experiment” map was generated was not sufficiently clear, and we regret that the choice of name created a fundamental misunderstanding of our method. Before describing the revisions, first let us address some apparent misunderstandings.

First, the halos are not removed by filtering. Instead, the GOODVIBES simulation is subtracted from the raw experimental map. GOODVIBES simulates all of the rigid-body motion in the crystal (as shown

previously in Meisburger, et al Nat Comm 2020), and it includes both acoustic phonons (halos) and optical phonons (cloudy scattering). Only after subtraction is a filter applied. The purpose of the (Savitsky-Golay) filter is not to remove halos (they have already been subtracted), but to smooth the residual (outliers are inevitable because halo scattering is so intense) and to interpolate onto a new grid of points (here, where h, k, l are integers).

Second, the MD simulation should be compared with the “corrected experiment”, not the “subtracted experiment”. The latter was included in Figure 6b in order to understand what aspects of the MD simulation contribute most to the correlation coefficient. To create an appropriate target for MD, we added back the diffuse scattering from GOODVIBES at the appropriate k -values for the unit cell simulation (here, $k=0$). In our one unit cell example, the scattering that is added back is not halo scattering (at $k=0$, there are no acoustic phonons), but the optical rigid-body modes of the unit cell.

We have extensively revised our presentation of the “corrected experiment” map to improve clarity. As detailed below, we have chosen a more precise name for the map, re-written the motivation and methods sections, simplified Figure 6, and added Supplementary Figure 6.

We agree that “corrected experiment” could be misunderstood. We now refer to this kind of map as a “target diffuse map” in the main text and the figures. We have added a clearer statement of our motivation for developing the target diffuse map to the main text on p. 6:

Here, we asked whether small-scale simulations, of one or a few unit cells, might be compared with the subtracted maps in order to maximize the proportion of the scattering signal from internal motion. A potential issue with such comparisons is that small-scale simulations still include external protein motion to some extent. Since GOODVIBES also involves a periodic supercell, we reasoned it could estimate the amount of external motion that ought to occur in an MD simulation of arbitrary size. This allowed us to create a “target diffuse map” for direct comparison with any particular crystalline MD simulation.

As you suggest, there might be other methods to achieve the same goal, however the target map has unique advantages. It is a straightforward application of GOODVIBES and we feel it increases the potential impact of the workflow.

The methods section pertaining to the target map (p. 24-25) was re-written to emphasize the motivation behind each step, and it is now illustrated using a flow chart and example data (Supporting Figure 6a).

We chose to simplify Figure 6 to focus attention on the comparison between MD simulation and the target map. The decomposition of the signal into “internal” (subtracted map) and “external” (GOODVIBES simulation) is still necessary for a thorough analysis, but is not central to the conclusions of the paper. We therefore moved this comparison to Supporting Figure 6b.

- [point 2] p. 12, line 267-268. “In our case, the strong agreement stems not from an improved force field per se, but from a more sophisticated data treatment.” The correlation of the MD with the “corrected experiment” is comparable to the “rigid body modes” below about 3 Å resolution, and both are substantially higher than the “subtracted experiment” in this range. At higher resolution, the correlations of the MD with the “subtracted experiment” and “rigid body modes” are comparable, and the correlation with the “corrected experiment” is increased compared to either alone. This means that most of the strong agreement with the “corrected experiment” comes from the GOODVIBES model and not from a more sophisticated data treatment (see above, the “corrected data” is not really the result of a data treatment, as it involves adding a model to the data).

The criticism of the “corrected experiment” appears to be related to a misunderstanding of how it was generated (see response to point 1). As discussed above, the “corrected data” involves subtracting and then adding back a model, and it has been renamed “target diffuse map” to better reflect its purpose and how it is made. The “subtracted experiment” is only the subtraction step where the GOODVIBES model has been fully subtracted. In this sense, the “corrected data” are less biased by the model.

- [point 3] p. 13, Fig. 6. The labels in the figure legends are a little confusing. According to the caption, the green symbols correspond to the filtered diffuse data (processed to remove the haloes). However, in the panel legend it is described as “Subtracted experiment” or “subtracted experimental map”. The blue corresponds to the GOODVIBES model fit to the haloes, but the caption just describes it as “Rigid body modes,” which doesn’t completely identify it as a model. See also previous comment about renaming “corrected experiment”.

As described above (in response to point 1), haloes are not filtered from the diffuse data. To reduce potential for confusion, we no longer include the “subtracted experiment” or “rigid body modes” in Figure 6. They have been moved to Supporting Figure 6b, and the labels are made consistent and clear in both figures and their captions.

- [point 4] p. 13, Fig. 6b, top panel. The agreement of the MD simulation with the isotropic data is remarkable. It is not yet perfect, however. The agreement of the isotropic component of the data with the MD is best at high res (above the peak) for the “subtracted experimental map”, and best at low res for the “corrected experiment”. Do the authors have any insights into why adding in the GOODVIBES model should increase the agreement at low resolution but decrease the agreement at high resolution? It would be nice to highlight this result a bit more in the manuscript, it’s impressive, even though there’s still room for improvement.

The most meaningful comparison is the “corrected experiment” (now target diffuse map) (see response to point 1 above). We agree that it is remarkable but not perfect and understanding the source of the discrepancy is of great interest for future work.

- [point 5] p. 21, line 511. It is not obvious that the cross terms between groups can be neglected. A more natural way of proceeding here would be to assume that the $V_{nkk'}$ are the same for all pairs of groups, so that $V_{nkk'} \rightarrow V_n$. With $V_{nkk'} \rightarrow V_n$ Eq. (9) becomes

$$I_D(\mathbf{q}) = N \sum_n e^{i\mathbf{q} \cdot \mathbf{R}_n} (\mathbf{q} \cdot \mathbf{V}_n \cdot \mathbf{q}) \sum_{\kappa, \kappa'} e^{i\mathbf{q} \cdot (\mathbf{r}_\kappa - \mathbf{r}_{\kappa'})} F_\kappa(\mathbf{q}) F_{\kappa'}^*(\mathbf{q}).$$

Note: The factor of N is missing in the authors’ equation, I believe this is needed to account for the reduction of the double sum to a single sum over lattice points. The Fourier transform of this equation immediately yields Eq. (14) in the manuscript for all space groups (again, up to a factor N), without further assumptions:

$$\begin{aligned} 3D-\Delta PDF(\mathbf{r}) &= N \sum_n \mathcal{F}\{\mathbf{q} \cdot \mathbf{V}_n \cdot \mathbf{q}\} * \delta(\mathbf{r} - \mathbf{R}_n) * P(\mathbf{r}) \\ &= N \sum_n \mathcal{F}\{\mathbf{q} \cdot \mathbf{V}_n \cdot \mathbf{q}\} * P(\mathbf{r} - \mathbf{R}_n) \end{aligned}$$

Here, $P(\mathbf{r})$ is the unit cell Patterson. Changing the math in this way would simplify the presentation, and connects the DISCOBALL model with the liquid-like motions model, with which it is closely related. The DISCOBALL model is distinguished from the liquid-like motions model in that it assumes that all pairs of atoms between the same two unit cells have the same covariance matrix of displacements (joint ADPs), whereas the liquid-like motions model assumes that the covariance matrix V is a continuously varying function of the separation between atoms in the crystal.

We agree that cross-terms between groups cannot be neglected if the goal is to reproduce the entire 3D-deltaPDF. However, DISCOBALL is only concerned with the peaks. The peaks are much more intense than the surrounding fluctuations due to cross-terms, and thus for our purposes the cross-terms can be neglected.

We also agree that making the assumption that all groups in the unit cell move as a rigid body ($V_{nkk'} \rightarrow V_n$) greatly simplifies the derivation of DISCOBALL. However, we purposefully avoided this assumption because the assignment of groups to particular unit cells is arbitrary for many space groups. For a concrete example, consider a crystal in P2 with a row of "d" shaped molecules: pdpdp... Are the first d and the second p in the same unit cell, or adjacent unit cells? Is the unit cell "pd" or "dp"? Both choices correspond to the same crystal, so the choice is arbitrary. Thus, we can't assume that $V_{nkk'}$ are the same for the same "n" when k and k' differ. DISCOBALL sidesteps the issue by focusing only on the peaks where $k=k'$ terms predominate, and there is no ambiguity about unit cell assignments.

To be consistent with previous work (especially our 2020 Nature Communications paper), the diffuse scattering is defined per unit cell (thus the factor of N has been divided out). The revised manuscript describes equation 9 as "diffuse intensity per unit cell" on p. 21.

As an aside, DISCOBALL is indeed similar to liquid-like model, but the scope is more limited. Liquid-like model aims to reproduce all of the 3D-deltaPDF. DISCOBALL aims to reproduce only certain regions (i.e., the peaks) of the 3D-deltaPDF where simplifying assumptions can be applied for the purpose of extracting displacement covariances for certain pairs. A complete deconvolution of the 3D-deltaPDF might be possible, but we do not attempt it here.

- [point 6] p. 21, last para. It's not clear what Patterson is used for the calculations in symmetric space groups. Is it the Patterson of the P1 unit cell, or is it a symmetrized Patterson computed just from the asymmetric unit? With the assumption $V_{nkk'} \rightarrow V_n$ mentioned above, the Patterson of the P1 unit cell would be well justified.

The Patterson we use is the standard one, i.e. the autocorrelation of the mean electron density of the crystal, or equivalently the folded autocorrelation of the mean unit cell electron density. To avoid confusion, in the main text we describe it as the "Fourier transform of the Bragg intensities" on p. 2.

- [point 7] p. 22, Eq. 15 – Each pair of equivalent groups in different unit cells can be superimposed on each other equivalent group via a translation and rotation of the crystal. Assuming any differences in the morphology of the crystal in different orientations can be neglected, all of the covariances between equivalent groups in different unit cells should, in principle, be the same.

It is true that the trace of V_{nkk} (the overall covariance) should be rotation invariant, and thus the same for all k . However, the challenge is to determine the off-diagonal elements from the symmetry-averaged peak (i.e. to invert Equation 13). It appears to us that this inversion is underdetermined in the most general case (i.e. when P_{kk} has approximate rotational symmetry near the origin: see response to point 8,

below). However, as we note in the paper, the inversion of Equation 13 may be possible if more information is included (such as long-r information in $P_{kk}(r)$).

- [point 8] p. 22, line 523. The different $P_{kk}(r)$ will only be similar up to a symmetry transformation.

We have clarified this point in the revised text on p. 21-22:

The $P_{kk}(r)$ are identical except for a rotation operator, and at small r , they contain a strong peak that is dominated by individual atom contributions. This peak region is expected to have approximate rotational symmetry except in special cases (such as when the resolution is highly anisotropic). Thus, in the general case, the peak does not contain enough information to deconvolve the contributions of each rigid group individually.

Summary

The major innovation of this paper is the DISCOBALL method, which enables lattice displacement covariance matrices (joint ADPs) V_n to be extracted in a data-driven way from the computed 3D- Δ PDF. Because they are able to fit multiple such matrices using the data, the amount of information the authors extract from the data using this method far exceeds that previously obtained using liquid-like motions (LLM) models, which assume a simple functional dependence of the covariance matrix on the separation between atoms. For example, Clarage et al (1992) used four parameters to describe isotropic haloes plus cloudy diffuse features in lysozyme, and Wall et al (1997) used 6 parameters (the model contained 12 but the principal axes were fixed, eliminating 6) to describe the diffuse streaks in calmodulin. Here the authors express each joint ADP using 6 variables; from the scatter plots in Fig. 4 there appear to be $O(102)$ joint ADPs that were fit (I cannot find the numbers, it would be good to provide these somewhere if not already in the ms), meaning that there might be more than 103 joint ADP variables that come out of the DISCOBALL fits for each dataset. This information provides a detailed picture of the lattice dynamics and should be useful both for gaining insight into the internal dynamics of the protein and for explaining all of the X-rays in the diffraction experiment, which have long been goals of protein diffuse scattering research.

The numbers of unique peaks fit by DISCOBALL (those in the asymmetric unit of the 3D-PDF) for each space group have been added to Supplementary Table 3. The numbers are 787 (triclinic), 84 (orthorhombic), and 36 (tetragonal). A reference to this table was added to the main text on p. 9, reproduced here for reference:

The number of unique joint-ADPs that can be determined by DISCOBALL analysis depends on the number of peaks in the ASU of the 3D- Δ PDF, which is a function of space group symmetry, unit cell size, and reciprocal space sampling (Supplementary Table 3).

The part of the manuscript that deals with the use of GOODVIBES to compare the MD simulations to the data is an interesting addition but needs some revision. I do believe the methods described will be able to help to extract information about internal motions, but right now it includes a comparison between the MD and a data-enhanced model that is instead being represented as a comparison between the MD and the data, along with a claim that the agreement is among the best reported previously for similar simulations. I understand what the authors are trying to do here, but I think this comparison needs to be presented in a different way to remove any confusion and to avoid misleading the reader. There might also be other comparisons that would demonstrate the ability to extract information about internal dynamics. For example, might it be possible to remove the lattice dynamics from the MD and then compare to the subtracted experiment? Or if the authors can somehow combine just the internal motions

of the MD with their GOODVIBES model and compare to the total diffuse signal, that might also prove to be a good approach. There are multiple ways of enhancing the presentation in this section — the most important thing is to increase the clarity and decrease confusion. There are currently only a couple of paragraphs in the results dedicated to the MD, and although it would be interesting to learn more here, it's not really necessary; the DISCOBALL/GOODVIBES results are very important and can carry the paper.

Michael Wall

We thank Michael Wall for his thoughtful comments and suggestions, which have significantly improved the clarity of our manuscript. We have renamed the map to which MD is being compared and have clarified how the generation of the target map was a natural progression of GOODVIBES analysis and how its purpose is to ultimately gain atomistic insight into protein motions. It is urgent for the field to take the next step from “understanding diffuse scattering” to “understanding proteins”; to accelerate this process, we have provided one tool (perhaps others are possible) to interface diffuse scattering analysis with MD.

Reviewer 2

The authors describe a significant breakthrough in the development of their X-ray crystal diffuse scatter analysis software. A key hurdle to this type of analysis is separating contributions from the "interesting" motions, such as functionally-relevant correlated movements within the molecule to the "uninteresting" diffuse scatter that comes from defects and other disorder in the crystal lattice itself. The authors make this distinction by declaring "lattice disorder" to be rigid-body movements of the molecules in the crystal. In this case there is only one peptide chain per enzyme, so the distinction is clear. Focusing on the halos that surround the Bragg spots, they fit the spring parameters in an elastic network model to a subset of these halo shapes with their "GOODVIBES" program, achieving an excellent match to all halos for not just one but three different crystal forms of lysozyme. This model then allows the "lattice disorder" component of the diffuse scatter to be simulated and then subtracted off the observed pattern, leaving behind the weaker but more "interesting" (to enzymologists) short-range correlated motion signal.

They also directly transform the halo data using a different approach called "DISCOBALL". This may not be quite as independent as the authors posit, since it is the same halos being either fit or transformed, but the algorithm is indeed radically different, lending confidence to the correctness of both implementations.

We thank the reviewer for the positive comments. We would like to clarify that although it is true that both methods fit the same dataset, they deal with different aspects of it. GOODVIBES fits a small fraction of the halos in a particular resolution range (typically 1-2%). DISCOBALL fits a Fourier transform of the entire diffuse map around particular regions in real space. These regions are related to the halos, but are not equivalent to them.

The impact of this achievement is potentially very exciting. Many enzymologists and others biologists with an interest in function will want to try this on their favorite systems. Although there are other diffuse scattering software packages that have been around for a while, these authors are neo-pioneers in the field, and have now solved many of the problems of data collection and data reduction for diffuse scatter X-rays. Here they demonstrate convincingly a key advance in data analysis, which is the separation of long- and short- range correlated motion. Even if the halos are being "over fit" by the admittedly complex underlying mathematical structure, this is still a good way to separate these two types of movement. I predict this will be a very high-impact paper.

A few specific comments.

Use of the word "subtle" to describe the correlated movements that make enzyme function possible might be inappropriately underwhelming. Perhaps a stronger word such as "key" or "critical" or "functional" would be better?

We thank the reviewer for the suggestions. In the abstract (p. 1), we replaced "subtle" with "key". Elsewhere, we have retained "subtle" because it underscores the fact that motions we're measuring can be extremely small (sub-angstrom) and difficult to detect by any other method.

"not yet been realized." - this is not the first diffuse-scatter software package. might be politic to mention the closest anyone has gotten? I recommend generally against the use of "priority language" as it is difficult to back up and ultimately has little archival value.

Thank you for this question. It is true that other software and approaches have been reported. Here, we meant that a complete workflow from data collection to understanding protein motions with detail has not been realized. To improve the clarity, we have revised the sentence on p. 2 as follows and cited our 2017 Chemical Reviews article where we comprehensively described previous efforts in the field:

However, although extensive efforts have been made in understanding protein diffuse scattering [Meisburger et al. 2017], a general workflow for utilizing this information has not yet been realized.

"accounted for entirely by physical models," - confusing - is it "entirely accounted" or "entirely physical" ? If the former, I don't think even the present work is accounting for everything, and if the latter, are previous formalisms like the "liquid-like-motions" really non-physical? This sentence needs to be better.

This sentence refers to our 2020 Nature Communications paper, which includes an elastic network model for internal motions that is absent in the present work. Liquid-like motions do not account for B-factor variations and is empirical rather than physically-motivated (although it has been shown that ENMs tend to have correlation functions resembling the liquid-like model). However, we agree that there is a strong potential for misunderstanding. We have revised text on p. 2 to be more precise:

[...] entirely and self-consistently described by physically-motivated atomistic models

"covariances for pairs of protein chains" - not atoms? Does not the covariance change from atom to atom? This threw me.

The text on p. 2 now reads "rigid-body displacement covariances for pairs of protein chains".

"absolute intensity" - please provide units at first mention

Units have been added to Fig. 1 caption.

"maximum tolerable dose was set conservatively" - please provide the actual dose in Gy (J/kg)

The dose was approximately 65 kGy per wedge. We added this number to the main text on p. 7, accompanied by a reference to the 2020 Nature Communications paper, where the dose estimate is discussed in greater detail.

Fig 3: "central sections (middle vs. bottom row)" Which is which? real vs sim? Suggest adding labels to figure

"corrected map" - unclear - although described more thoroughly in the extended section, the description in the main text is ... hard to follow. This reviewer had to read it many times before really understanding what the "corrected map" is. And even then: what filter? In cases such as this a statement as to the motivation for the "corrected map" can be helpful. Such as "we added back the lattice-disorder DS to the MD-derived DS for a more direct comparison to the experimentally observed DS". Or something along those lines.

We thank the reviewer for raising this point, which was also raised by Reviewer 1. As described in greater detail in response to Reviewer 1's point 1, we have revised the text by renaming "corrected map" to "target diffuse map" to better reflect its purpose and how it was made and revised the main text, methods, Figure 6, and Supplementary Figure 6.

"hen egg white" - ambiguous - replace with species name. I.E. an adult female *Meleagris gallopavo* is called a "hen", but I suspect the protein used here was from *Gallus gallus* ?

We have revised the methods on p. 15 to include the species name, *Gallus gallus*.

experimental amplitudes - why not use 2mFobs-DFcalc map? This map is the closest to the "true" electron density. Much closer than an Fo map phased with PHCalc. Although it is a clever advance to use Fobs, I predict that using the standard sigma-A weighted maps will be better. Might need to be scaled to be on the same scale as Fcalc, but otherwise should be compatible with the algorithm as described. Note: I do not present this as a requirement for publication, but merely a suggestion to the authors. Should they go ahead with the Fobs@PHCalc density, however, perhaps a sentence as to why the 2mFobs-DFcalc map was not used would be in order.

It is true that the sigmaA-weighted maps should be closer to the 'true' electron density than Fo phased with PHC. For our purposes, it is more important to have an accurate molecular transform, especially near the Bragg peaks where the halos are most intense. Fo phased by PHC seemed like a good way to achieve this. However, we did not extensively test alternatives, and 2mFo-DFcalc might have certain advantages. We thank the reviewer for the suggestion.

"averaging the averaging the electron density" - duplicate phrase

Fixed, thank you!

"the value of the intensity and its uncertainty were computed at the target grid location"
would it be more accurate to say the intensity and uncertainty were interpolated to the target grid? Or perhaps I misunderstand?

In the Savitsky-Golay method, each neighborhood of data points is fit by a polynomial (of order less than number of points) and the value of the polynomial is computed at the target location (typically the center of the neighborhood). Experimental errors are propagated to reflect the uncertainty of the fit at the target location. It is distinct from polynomial or spline interpolation, in that the fitted curve typically does not pass through the data points exactly (this makes it a filter rather than an interpolant). We modified the text on p. 24 to make this clearer.

For each neighborhood, a second-order polynomial was fit using weighted least-squares [Press et al. 2007]. Then, the value of the polynomial was computed at the target grid location, and the corresponding uncertainty was propagated from the experimental errors.

Extinction? or pile-up? The non-linear disagreement between F_{obs} and F_{calc} noted in the extended figures could be due to something other than extinction. With crystals of this size, extinction would be surprising. An alternative and testable hypothesis is that the low mosaic spread of these room-temperature crystals is resulting in very narrow rocking widths that overwhelm the detector's maximum count rate. For the Pilatus3 used here this is $1e7$ photons/s/pixel. The "seconds" used here are not the exposure time, but rather the time it takes the beam to transit the Ewald sphere, which can be very short for narrow mosaic crystals in low-divergence beams. Since the authors are working on the absolute photon scale already, it should be a simple matter to check how close the instantaneous maximum count rate came to the detector's rated maximum. Yes, Pilatus3 has the "instant retrigger" feature, but residual non-linearity can still happen despite this correction. Alternatively, the observation of this non-linearity does not contribute to the main thrust of this paper, and it would be equally impactful (and shorter) if it were left out.

We agree that for the purposes of this paper it does not matter whether reduction of Bragg intensities at low resolution is caused by extinction or pile-up. The important thing is that a simple correction (the extinction correction) can rescue the situation. In our case, the peak count rates do approach the regime where count-rate corrections kick in, so it is a plausible explanation. It is straightforward to distinguish between pile-up and extinction experimentally (e.g. by attenuating the beam), and it is something we are pursuing in future work. However, answering the question is beyond the scope of this paper.

We have modified the main text on p. 6 as follows:

When investigating the accuracy of our Bragg data, we unexpectedly found that intensities were systematically suppressed in an intensity- and resolution-dependent manner. Interestingly, this behavior is reminiscent of dynamical scattering (extinction), and we corrected for it using a method developed for small-molecule crystallography (Supplementary Fig. 2).

In the caption to Supplementary Fig. 2, we added the following text:

Detector count-rate artifacts [Trueb et al. 2015] might also explain the suppressed Bragg intensities, and cannot be ruled out without further experiments.

Well done all, and I look forward to the final version in print!

Reviewer 3

Short

The paper introduces methodology for fitting diffuse halos around Bragg peaks of protein crystals. It addresses an important problem of using single crystal diffuse scattering for understanding the protein dynamics. Diffuse scattering is a continuous signal which is always present alongside Bragg peaks, and presents information about variability in the structure. In the case of proteins, diffuse scattering probes contain conformations that are dynamically accessible for the crystallized protein which in many cases is related to the protein function. Unfortunately, due to complexity of modeling diffuse scattering, information which it contains is currently ignored.

The paper presents an important progress to solving this problem. It presents two methods for fitting a portion of diffuse scattering which is present in a form of halos around Bragg peaks. The methodology is general and applicable to most of the proteins. It is implemented in well-written and well-documented scripts and in my view has a great potential to be widely adopted in analysis of protein diffuse scattering.

I believe that the presented methods will have a major impact, and I recommend publishing the paper with minor corrections.

We thank the reviewer for the positive comments.

Minor comments

Page 10, figure 4a: Would it be possible to add a faint gray line at the covariance = 0 \AA^2 in order to visually highlight correlations approaching negative values for long pairs? Please note that in my view the negative values of the correlations are completely justified and their origin is well explained in the text (page 7 198-211).

Faint gray lines where y-axis = 0 have been added to Fig. 4a.

Page 31, Extended figure 4: Colormap is inverted (brown-negative, purple - positive) wrt colormap 5b. Could you flip the colors in the figure 4?

Thank you for pointing out that different colormap conventions were used in main text and supporting figures. We flipped the colormap in Supplementary Figure 4 to match the main text Figure 5.

Also what are the units that you use in figure 4? Shouldn't autocorrelation be expressed in electrons squared per volume (e^2/vol). Please note that in my view it is not important to find the absolute scale, relative scale (say $P/\max(P)$ where P is the 3D- Δ PDF is sufficient in this case.

The 3D- Δ PDF is on an absolute scale, typically divided by the number of asymmetric units so that different crystal forms are comparable. We modified legends in Supplementary Figure 4 and Figure 5 to make this clearer and added physical units as suggested.

Reviewers' Comments:

Reviewer #1:

Remarks to the Author:

Please see the attached .pdf

Reviewer #2:

Remarks to the Author:

The revised manuscript is a significant improvement, and now much clearer. The authors have addressed all concerns and issues I was able to identify. I believe this manuscript is now ready for publication!

Please find enclosed the second revision of our manuscript, “Robust total X-ray scattering workflow to study correlated motion of proteins in crystals.” Reviewer 1’s comments are reproduced verbatim and addressed point-by-point below. For clarity, reviewer comments are in black and our response is in blue. Revisions to the manuscript are highlighted in yellow.

Reviewer 1

The authors correctly surmised that there was a miscommunication concerning the target diffuse map (formerly corrected experiment). My original concerns have been addressed by the revision, the clarifications helped a great deal in this regard.

Most of my other comments have been addressed in the revision. An exception is the DISCOBALL derivation, however. One concern I had raised is that it isn’t obvious that the cross terms between groups can be ignored. In the response the authors address this by stating “The peaks are much more intense than the surrounding fluctuations due to cross-terms, and thus for our purposes the cross-terms can be neglected.” Unfortunately this doesn’t adequately address the concern, as it still doesn’t explain why the cross terms can be ignored in the peaks. Perhaps they are thinking that these terms aren’t important because they involve long-distance contributions to the Patterson, and that the peaks are relatively sharp. Indeed, at long distances, the inter-molecular atom pairs can dominate the Patterson; however, even at modest distances, the Patterson still can include contributions from a substantial number of atom pairs that cross molecular boundaries. Moreover, Fig. 2c shows that the DISCOBALL model does include contributions at distances that are a substantial fraction of the unit cell dimension. If the authors wish to use this assumption in their derivation, it will be necessary for them to provide evidence that it is reasonable, e.g., by computing $P(r)$ either way and comparing the results, and/or including information about the proportion of contributions from atom pairs within vs across groups at various distances.

We agree that, in general, the cross-terms between rigid groups are significant at modest distances, and must be considered when calculating diffuse scattering accurately from a model. To be clear, the GOODVIBES simulations do include these cross-terms, as well as correlated rotational motion of the rigid bodies. In DISCOBALL, we can neglect such contributions because we only need to model the 3D- Δ PDF at the lattice points. The peaks at the lattice points represent the constructive interference of scattering from atoms separated by a lattice translation. Because there are many such pairs of atoms, the peaks at the lattice points are very intense, and they dominate the signal when performing deconvolution. To be certain that only the lattice peak signal contributes to deconvolution, we applied a spherical mask with a radius of 4 Angstrom to each extracted peak. In Figure 2c, the extracted peaks are shown in the upper right panel -- we would not consider the orange box to be a substantial fraction of the unit cell. Regardless, we are happy to provide evidence that cross-terms can be neglected in DISCOBALL.

We have added a figure (Supplementary Fig. 9b) that compares the 3D- Δ PDF peaks for tetragonal lysozyme computed exactly (using GOODVIBES) with the corresponding DISCOBALL approximations using the ground truth values for the joint ADPs (equations 14 and 15). There are subtle differences between the two that result from various effects (including neglect of rotations, cross terms, symmetry averaging, and different numerical algorithms). However, overall DISCOBALL does an excellent job capturing the central peak shape, which confirms that these effects can be safely neglected for the purposes of deconvolution. This figure is referenced in Methods section of the main text on page 22, which has also been revised to improve clarity (reproduced below):

Instead, we assume that the peak can be approximated by its symmetry average, or equivalently the Patterson peak for the unit cell (Supplementary Fig. 9a, middle and right panels). Then, the peak shape is described by Eq. 14 with an effective joint-ADP that is equal to the average over groups, as follows:

[Eq. 15]

Modeling the symmetric crystal as though it were P1 involves assumptions that are not obviously valid. However, lattice dynamics simulations show that the error incurred is small enough to be neglected in the cases studied here (Supplementary Fig. 9b).

I also don't think the authors' reason for rejecting the simplified derivation holds up to further analysis. They stated in the response that they wished to avoid the key assumption $V_{nkk'} \rightarrow V_n$ because the assignment of groups to unit cells is arbitrary. However, they already make this assumption later in Eq. (15).

We do not make the assumption that $V_{nkk'} \rightarrow V_n$ in eq. 15. Rather, we define V_{eff} as an average over $V_{nkk'}$. V_{eff} is then substituted into Eq 14. The substitution is valid if either of two things are true: (a) $V_{nkk'} = V_n$, as you note above, or (b) $P_{kk}(r)$ is invariant to symmetry transformations at small r . We have argued in the manuscript that (b) is approximately true when resolution is isotropic, and so it is not necessary to make assumption (a). We have added a numerical illustration of this in Supporting Fig. 9a, where $P_{kk}(r)$ of the tetragonal lysozyme asymmetric unit is compared with its symmetry average and with the unit cell $P(r)$. The central peak (red color in Fig. 9a) is approximately the same in all cases. The reason we do not assume (a) and instead rely on (b) is because Eq. 15 gives a recipe for simulating peaks from an atomistic model where (a) is not true (such as GOODVIBES). For instance, we use Eq. 15 to compare GOODVIBES and DISCOBALL joint ADPs in Fig. 4.

Making this assumption earlier leads to the same equations the authors are using but without the additional assumption that the cross terms must be ignored. In addition, as long as the unit cell structure factor is represented using DFTs, the arbitrary assignment of groups to unit cells should not influence the result, due to the periodicity property. This can be seen by inspecting the following equation

$$I_D(\mathbf{q}) = N \sum_n e^{i\mathbf{q} \cdot \mathbf{R}_n} (\mathbf{q} \cdot \mathbf{V}_n \cdot \mathbf{q}) \sum_{\kappa, \kappa'} e^{i\mathbf{q} \cdot (\mathbf{r}_\kappa - \mathbf{r}_{\kappa'})} F_\kappa(\mathbf{q}) F_{\kappa'}^*(\mathbf{q})$$

The double sum over kk' is just the squared structure factor of the unit cell, which is the same no matter how the unit cell is defined, so long as it is complete. The choice of how to make the assignment therefore doesn't influence the result, at least if DFTs are being used. I do encourage the authors opt for the simplified derivation, which eliminates the need to assume the cross-terms are negligible; however, provided the authors can present evidence that the cross-terms actually are negligible, the simplified derivation isn't required.

As noted above, we now present evidence in Supplementary Fig. 9 to demonstrate that the DISCOBALL assumptions work in practice. However, we appreciate the point that DISCOBALL can be derived with a different set of assumptions, and that DISCOBALL is similar from a mathematical point of view to the liquid-like model (LLM), which you discussed in depth in your previous review. The LLM has been one of the most successful phenomenological models for protein diffuse scattering, and thus we agree that it would be helpful to the field for us to acknowledge the mathematical similarity between DISCOBALL and LLM and explain how they differ. We have therefore added the following text to the manuscript (page 22):

Finally, we note that the DISCOBALL formula (Eq. 14) resembles the liquid-like model (LLM) for diffuse scattering [Caspar et al. 1988, Meisburger et al. 2017, and Wall et al. 1997]. The LLM is derived by assuming that joint ADPs depend only on the inter-atomic vector. We do not make that assumption explicitly when deriving DISCOBALL, however doing so has certain advantages; it would ensure that Eq. 14 applies regardless of space group symmetry, and that Eq. 14 remains valid for the entire 3D- Δ PDF, not just the peak regions. Our derivation, though more complex, uses a more limited set of assumptions so that joint ADPs are directly comparable with those derived from atomistic models such as GOODVIBES.